# Decoding non-coding SNPs: systems genomics modelling dissects the heterogeneity of IBD

Dezső Módos [1,2,3,4,5,18], John P Thomas[2,6,18], Johanne Brooks-Warburton[3,4,7,8,18], Martina Poletti [3,4], Balazs Bohar[2,4,9,10], Yufan Liu [2], Matthew Madgwick[3,4,11], Luca Csabai[9], Wen-Xin Kang[2], Benjamin Alexander-Dann[5], Azedine Zoufir [5], Padhmanand Sudhakar[4,12], Domenico Cozzetto[2], David Fazekas [4,9], Shamith Samarajiwa[1], Simon R Carding [3,13], Nicholas Powell[2], Bram Verstockt[12,14], Andreas Bender[5,15,16,17] & Tamas Korcsmaros [2,3,4 ✉]

## Abstract

Genome-wide association studies have identified numerous susceptibility loci in complex diseases, such as chronic immune-mediated inflammatory disorders (IMIDs), yet their impact on pathomechanisms remains poorly understood. Low effect sizes, polygenicity, and predominance within non-coding genomic regions remain major challenges to the functional interpretation of IMID-associated single-nucleotide polymorphisms (SNPs). To address this, we present a novel systems genomics approach which models the cumulative impact of non-coding SNPs on downstream cellular signalling and gene regulatory networks. Applying this to the prototypical chronic IMIDs of Crohn's disease (CD) and ulcerative colitis (UC), both forms of inflammatory bowel disease (IBD), we individually analysed 2,636 patient genomes. Signals from non-coding SNPs were found to propagate towards well-established and novel CD- and UC-associated pathogenic pathways through the signalling and gene regulatory layers. The SNP-propagated gene regulatory networks stratified CD and UC patients into distinct clusters corresponding to cell type-specific gene dysregulation and potential therapeutic response. This approach bridges the gap between genotype and phenotype, laying the foundations for accelerating precision medicine in complex diseases.

Keywords Precision Medicine; Single-nucleotide Polymorphisms; Systems Genomics; Network Propagation; Inflammatory Bowel Disease
Subject Categories Digestive System; Genetics, Gene Therapy & Genetic Disease; Immunology

## Introduction

Following the sequencing of the human genome at the turn of the 21st century, genome-wide association studies (GWAS) have sought to identify the genomic loci underpinning the heritability of common, complex traits and diseases (Visscher et al, 2017). These efforts have uncovered hundreds of susceptibility loci in chronic immune-mediated inflammatory disorders (IMIDs) such as rheumatoid arthritis and inflammatory bowel disease (IBD) (Parkes et al, 2013). Despite initial optimism that this would translate into a deeper understanding of disease pathogenesis and facilitate precision medicine strategies, this has yet to be realised (Joyner and Paneth, 2019).

A central challenge is that the majority of risk loci associated with complex diseases reside within non-coding regions of the genome, complicating their functional interpretation (Jostins et al, 2012; Stankey and Lee, 2023). Moreover, individual risk variants typically have small effect sizes and account for only a fraction of total disease heritability (Gordon et al, 2015), resulting in significant "missing heritability" (Manolio et al, 2009; Boyle et al, 2017). Accumulating evidence suggests that this missing heritability can in part be explained by the impact of non-coding genetic variants on gene regulation, particularly the expression of *cis*-linked genes (Maurano et al, 2012; Gusev et al, 2014)—a phenomenon recently termed "missing regulation" (Connally et al, 2022). Further

[1]Division of Systems Medicine, Department of Metabolism, Digestion and Reproduction, Imperial College London, London, UK. [2]Division of Digestive Diseases, Department of Metabolism, Digestion and Reproduction, Imperial College London, London, UK. [3]Gut Microbes and Health Programme, Quadram Institute Bioscience, Norwich Research Park, Norwich, UK. [4]Earlham Institute, Norwich Research Park, Norwich, UK. [5]Centre for Molecular Science Informatics, Department of Chemistry, University of Cambridge, Cambridge, UK. [6]UKRI MRC Laboratory of Medical Sciences, Hammersmith Hospital Campus, London, UK. [7]Department of Clinical, Pharmaceutical and Biological Sciences, University of Hertfordshire, Hertford, UK. [8]Department of Gastroenterology, Lister Hospital, Stevenage, UK. [9]Department of Genetics, Eötvös Loránd University, Budapest, Hungary. [10]Synthetic and Systems Biology Unit, Institute of Biochemistry, Biological Research Center, Szeged, Hungary. [11]IBM Research Europe, The Hartree Centre, Daresbury, UK. [12]Department of Chronic Diseases and Metabolism, KU Leuven, Leuven, Belgium. [13]Norwich Medical School, University of East Anglia, Norwich, UK. [14]University Hospitals Leuven, Department of Gastroenterology and Hepatology, KU Leuven, Leuven, Belgium. [15]College of Medicine and Health Sciences, Khalifa University of Science and Technology, Abu Dhabi, UAE. [16]STAR-UBB Institute, Babeș-Bolyai University, Cluj-Napoca, Romania. [17]Research Center for Functional Genomics, Biomedicine and Translational Medicine, Iuliu Hatieganu University of Medicine and Pharmacy, Cluj-Napoca, Romania. [18]These authors contributed equally: Dezső Módos, John P Thomas, Johanne Brooks-Warburton. ✉E-mail: t.korcsmaros@imperial.ac.uk

complexity arises from polygenic effects, in which multiple susceptibility loci interact to result in subtle perturbations in interconnected biological pathways that drive disease phenotypes (Mackay, 2014). Therefore, traditional experimental approaches that interrogate the impact of individual susceptibility loci on phenotype are unlikely to fully capture the polygenic and network-level disruptions that underlie complex diseases, such as IBD (Edwards et al, 2013; Rao et al, 2021). This highlights the need for integrative approaches capable of modelling the collective influence of multiple non-coding variants on gene regulation, gene expression, and, ultimately, phenotype (Connally et al, 2022).

Systems genomics has recently emerged as a powerful framework to address this need. By applying network biology principles, it enables the detection of salient molecular interactions within complex biological systems to elucidate gene-to-disease associations. Early approaches in the field used guilt-by-association principles to identify the direct interactors and local neighbourhoods of seed genes within biological networks to infer biological pathways associated with a disease (Schwikowski et al, 2000; Cowen et al, 2017). For instance, we previously generated the integrated Single-Nucleotide Polymorphism (iSNP) network pipeline (Brooks-Warburton et al, 2022) to predict the combinatory effects of non-coding single-nucleotide polymorphisms (SNPs) located within miRNA target sites and transcription factor binding sites (TFBSs) in promoter and enhancer regions by identifying SNP-affected proteins and their first neighbour interactors within the human signalling network. While informative, such local neighbourhood methods fail to capture the effect of genes or proteins on more distant connections or across biological layers, and thus do not fully contextualise seed genes or proteins within the global topology of biological networks (Badkas et al, 2022; Vanunu et al, 2010). Recently, we also applied the iSNP pipeline to analyse the upstream effects of SNPs located within TFBSs (Bohar et al, 2025). While this study introduced a novel approach for linking IBD-associated genotypes with upstream extrinsic and environmental signals, its analysis was limited to transcription factors whose binding was affected by these SNPs and their subsequent functional annotation (Fiocchi and Iliopoulos, 2025).

To capture the systemic and downstream effect of perturbations, including genetic alterations, novel methods based on network propagation have emerged that amplify biological signals across the global architecture of a molecular network (Cowen et al, 2017). In network propagation, seed genes or proteins are considered as "hot spots" within the network and their "heat" diffuses to identify other "hot" candidates across the entire network through an iterative process (Badkas et al, 2022; Cowen et al, 2017). Importantly, network propagation has been found to outperform local neighbourhood methods to infer gene-to-disease associations, largely from studies in the field of oncology (Cowen et al, 2017). Recently, random walk-based network propagation was employed to expand the known network of pleiotropic GWAS signals across multiple human traits, including chronic IMIDs, to reveal novel candidate genes and drug targets (Han et al, 2023; Seal et al, 2015). However, the application of such approaches in real-world patient cohorts remains unexplored.

Building on these concepts, here we present a novel patient-specific systems genomics framework that integrates individual genotype data with multilayered biological networks to model the cumulative effect of disease-associated non-coding SNPs on disease pathomechanisms. Unlike prior methods, we extend beyond protein–protein signalling interactions to also include gene regulatory networks, enabling the functional propagation of non-coding SNP effects to downstream transcriptional regulators that drive disease phenotypes. We illustrate the utility of this approach by performing a patient-level analysis of genotype data from 2636 individuals with Crohn's disease (CD) and ulcerative colitis (UC), which represent two distinct but related chronic IMIDs that fall under the umbrella of IBD (Huang et al, 2017).

Using this workflow, we show how signals from established disease-associated non-coding SNPs cumulatively propagate across cellular signalling and gene regulatory networks towards key biological pathways considered to be pathognomonic to CD and UC. Importantly, SNP-propagated gene regulatory networks, unlike SNP-propagated signalling networks or SNP-affected proteins, stratify patients into distinct molecular subgroups corresponding to cell type-specific gene dysregulation and potential differential response to biologic therapy. This work demonstrates the power of patient-specific systems genomics modelling across multilayered biological networks to bridge the elusive gap between genotype and phenotype in CD and UC, providing a novel paradigm for understanding complex genetic disorders and informing precision medicine in IBD and beyond.

## Results

### Generation of patient-specific SNP-propagated signalling networks and gene regulatory networks

We used network propagation to model how disease-associated non-coding SNPs impact downstream cellular signalling and gene regulatory networks. We contextualised this analysis to the prototypical chronic IMIDs of UC and CD by utilising genotype data from a large cohort of CD ($n = 1695$) and UC patients ($n = 941$) from the Leuven IBD Center as the input for the workflow (University Hospitals Leuven, KU Leuven, Belgium, Appendix Table S1). The workflow, which is disease agnostic, is summarised in Fig. 1.

Briefly, we first identified SNPs present in each patient genome in our cohort that are known to be associated with IBD based on landmark genome-wide association studies (Jostins et al, 2012; Farh et al, 2015) (Fig. 1A). We then filtered for IBD-associated SNPs located within non-coding genomic regions, including enhancers, promoters, or miRNA-target sites (Fig. 1B). Next, we employed complementary in silico tools (see "Methods") to identify non-coding IBD-associated SNPs predicted to disrupt transcription factor (TF) binding to TFBSs within promoters and enhancers, or alter microRNA binding to microRNA-target sites. Predictions were restricted to SNPs resulting in a gain or loss of a TFBS or miRNA-target site relative to the ancestral allele. To ensure tissue relevance in the context of IBD, we integrated chromatin immunoprecipitation sequencing (ChIP-seq) data and histone modification data from intestinal tissues, drawing from the ENCODE database (Qu and Fang, 2013), the ReMap database (Hammal et al, 2022), and a published dataset from suspected IBD patients (Mokry et al, 2014), enabling us to retain only those TFBSs located within epigenetically active promoter or enhancer regions. In this way, a gene and its translated protein were considered "SNP-affected" if a non-coding IBD-associated SNP was predicted

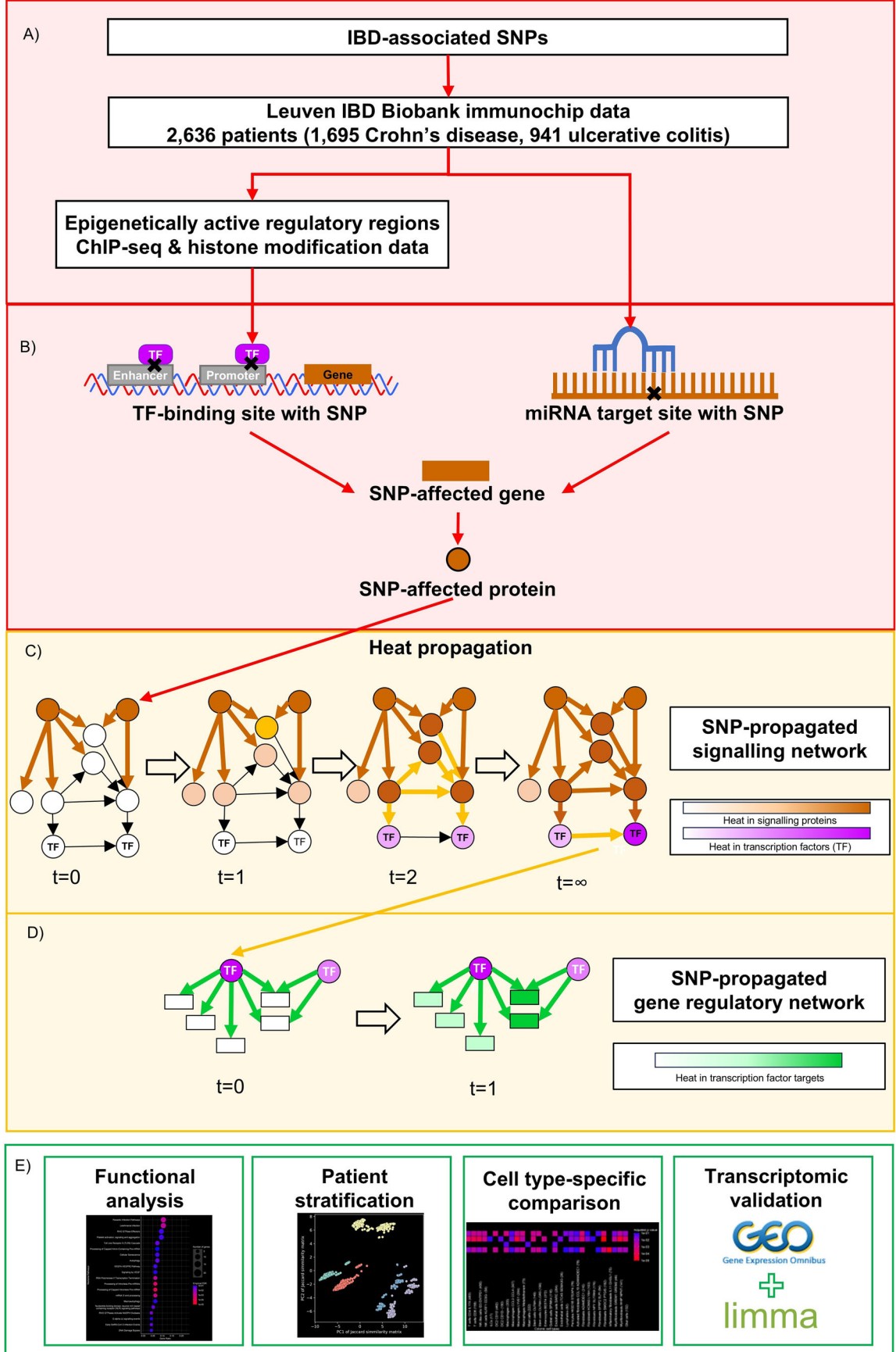

◀ Figure 1. Workflow of the current systems genomics study.

(A) We first analysed genotype data from 2636 IBD patients (1695 Crohn's disease, 941 ulcerative colitis) to evaluate the impact of established non-coding IBD-associated SNPs on transcription factor (TF) binding to TF-binding sites (TFBSs) within promoters and enhancers, or microRNA binding to microRNA-target sites. (B) A gene and its translated protein were considered "SNP-affected" if a non-coding IBD-associated SNP was predicted to modulate its expression via disruption of an epigenetically active TFBS or a microRNA-target site (Brooks-Warburton et al, 2022). (C) Using network heat propagation, we determined the impact of non-coding SNPs beyond local first-neighbour SNP-affected proteins to downstream proteins, including TFs, to generate a SNP-propagated signalling network. This allowed us to better capture the impact of non-coding SNPs on the global intracellular signalling network. (D) We then performed a further iteration of network heat propagation from the identified downstream TFs, to map the impact of non-coding SNPs on gene regulatory interactions. We defined this as the SNP-propagated gene regulatory network. (E) Using the SNP-propagated signalling networks and gene regulatory networks, we performed various analyses, including functional analysis, patient stratification, cell type-specific gene dysregulation comparison, and validation using transcriptomics datasets. t pseudotime, TF transcription factor. Source data are available online for this figure.

to modulate its expression via disruption of an epigenetically active TFBS or a microRNA-target site (Dataset EV1; Fig. 1B).

These SNP-affected proteins were then assigned as "seed nodes" for heat-based network propagation, which was run to convergence to identify downstream nodes likely to be perturbed within the global directed human intracellular signalling network (Türei et al, 2021) (Fig. 1C).

To characterise the biological processes impacted in this "SNP-propagated signalling network", we identified and annotated functional network units called modules. A network module is a part of the network which is more connected to its members than to other parts of the network, often representing particular biological functions.

To extend this signal through the downstream gene regulatory layer, we performed a second round of network propagation from TFs identified in the SNP-propagated signalling network to their target genes (Garcia-Alonso et al, 2019) (Fig. 1D). This "SNP-propagated gene regulatory network" was constructed using one-step heat propagation to reflect the broader, more promiscuous interactions that are known to occur between TFs and their targets compared to signalling proteins (Cowen et al, 2017). The resulting network formed modules centred around key TFs connected to target genes whose expression may be ultimately influenced by disease-associated non-coding SNPs. Finally, we annotated these gene regulatory modules to identify biological pathways and processes that could be perturbed in UC and CD through the propagation of non-coding SNP signals across signalling and regulatory layers (Fig. 1E).

In summary, this two-step, compartmentalised network propagation approach, enabled us to model polygenic contributions by integrating the impact of multiple disease-associated non-coding SNPs on downstream signalling and gene regulatory networks. A summary of the features (non-coding SNPs, SNP-affected genes, and SNP-propagated genes) per propagation step for both CD and UC is provided in Appendix Fig. S1 for all networks. To minimise random noise and the overrepresentation of hub nodes, nodes in both SNP-propagated signalling and gene regulatory networks were retained only if they passed a threshold of statistical significance (Z-score > 2) compared to a null distribution of 10,000 control networks generated from random seeds (see "Methods").

Leveraging these high-confidence network models, we next performed patient-specific analyses. Rather than aggregating all UC or CD-associated SNPs present in our cohort, we analysed each patient in turn based on their unique SNP profile. This allowed us to enrich and contextualise non-coding genomic risk variants to intracellular processes that may contribute to IBD-relevant pathomechanisms at the level of the individual patient.

## Non-coding SNPs in CD impact downstream signalling processes involving key immune pathways, DNA repair, and autophagy

We found 38 SNP-affected proteins in our cohort of 1695 CD patients (Dataset EV1). Using these proteins as seeds for the heat diffusion algorithm, we reconstructed a SNP-propagated signalling network comprising 82 proteins (nodes) and 208 signalling interactions (edges). This network consisted of four major modules (Fig. 2A; Dataset EV2) encompassing Reactome pathways related to cytokine signalling (in particular IL2 signalling), immune pathways (MAPK activation, TLR signalling, IL17 signalling), DNA repair, and PTEN regulation. In addition, smaller modules were identified that enriched for various pathways (Dataset EV2), such as gap junction degradation mediated by key autophagy genes. These findings reveal how the genetic background of patients may influence major signalling processes that are known to be involved in CD pathogenesis (Roda et al, 2020).

Non-coding SNPs were predicted to impact 68 proteins in more than 100 CD patients (Fig. 2B; Dataset EV3). These proteins were most strongly enriched for "signalling by interleukins" in Reactome (Jassal et al, 2020) (Fig. 2C), revealing a likely genetic basis for dysregulated interleukin signalling, which is a hallmark of CD (Neurath, 2014). Other significantly enriched pathways included those related to toll-like receptor (TLR) signalling (TLR2, TLR3, TR10, MyD88 activation) and parasitic infection. These findings suggest that mucosal responses to microbial stimuli (viral, bacterial and/or parasitic) may be frequently dysregulated in CD patients. Additional putative perturbed signalling pathways included those relating to DNA repair and growth factor receptor signalling. The commonly perturbed proteins did not significantly overlap with differentially expressed genes between CD patients and non-IBD controls from previous transcriptomics studies (Benjamini–Hochberg adjusted $P > 0.05$, Appendix Table S2).

## Patient-specific SNP-propagated gene regulatory networks stratify CD patients into five clusters

We hypothesised that propagating the signal further through a downstream transcription factor-target gene (TF-TG) regulatory network may better capture disease-relevant changes in gene expression. The CD SNP-propagated gene regulatory network comprised 1278 genes in total of which 8 were TFs, 1268 were target genes, and 2 were both TFs and target genes. These genes formed 1598 TF-TG interactions. The TF-TG regulatory network yielded six gene regulatory modules overall (Fig. 3A) corresponding to various CD-associated functions: (i) interleukin signalling, in particular IL4, IL12, 1L13 and IL20 (regulated by STAT3), (ii) extracellular matrix organisation and TGFβ signalling

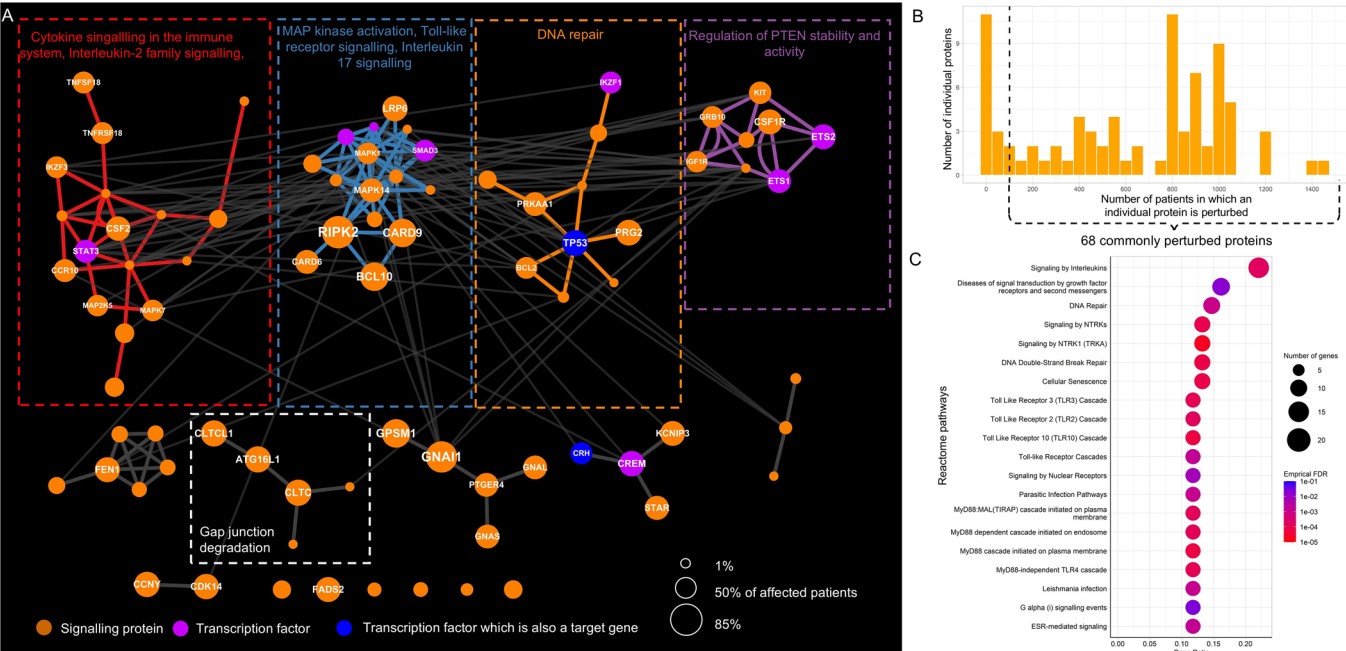

**Figure 2. Crohn's disease-associated non-coding SNPs specifically perturb cytokine signalling, immune pathways, DNA repair, and apoptosis regulation.**

(A) SNP-propagated signalling network in Crohn's disease (CD) with proteins (nodes) grouped by Girvan–Newman modularisation. The largest four modules were functionally annotated using Reactome enrichment. An additional smaller module ("gap junction degradation" involving autophagy proteins) has also been annotated. These modules were emphasised by adding coloured edges. Node size corresponds to the percentage of patients having the SNP-propagated protein. (B) Frequency distribution of SNP-propagated proteins in the CD patient cohort. (C) The enriched Reactome pathways of the most commonly perturbed SNP-propagated proteins in the CD patient cohort. Commonly perturbed proteins are those that are affected in more than 100 patients in the cohort. Source data are available online for this figure.

(regulated by SMAD3 and NR3C1), (iii) GPCR signalling (regulated by CREB1 and CREM), and (iv) CD28 co-stimulation (regulated by IKZF1). Two additional modules regulated by master regulators ETS1 and ETS2, and p53, respectively, were also identified but not functionally annotated (Dataset EV2). These findings reveal that disease-associated non-coding SNPs converge on major cellular pathways involved in CD pathogenesis, regulated by key transcriptional regulators.

In total, 980 target genes in the CD SNP-propagated gene regulatory network were present in more than 100 patients, forming two peaks representing STAT3 and TP53 targets (Fig. 3B). These commonly perturbed target genes significantly overlapped with CD-specific differentially expressed genes in the ileum and rectum (*P* < 0.05 Benjamini–Hochberg corrected *P* value, Appendix Table S2), and were functionally enriched for interleukin signalling (including IL4 and IL13 signalling), p53-related processes, and neutrophil degranulation (Fig. 3C; Dataset EV3). The recurrence of interleukin signalling and DNA damage repair pathways at both the signalling and regulatory layers suggests that SNP-propagated pathogenic feedback loops may connect the cellular signalling and gene regulatory layers of these pathways in CD. The enrichment of neutrophil degranulation, a hallmark of active IBD (Segal, 2018) and the basis of faecal calprotectin testing as a non-invasive biomarker (Jukic et al, 2021), indicates a potential genetic mechanism contributing to this process.

We next evaluated whether the SNP-propagated networks could dissect patient heterogeneity in our cohort through clustering analysis. The clustering of SNP-propagated gene regulatory networks yielded a higher average silhouette score compared to SNP-propagated

signalling networks and SNP-affected proteins, suggesting that patient heterogeneity is most effectively captured at the gene regulatory layer (Appendix Fig. S2A). Using k-means clustering, the SNP-propagated gene regulatory networks stratified the CD cohort into five distinct patient-specific clusters (Fig. 3D; Appendix Fig. S2A). While cluster definition was guided by quantitative metrics, it inevitably carries some subjectivity. Each patient cluster was characterised by a corresponding set of gene regulatory modules driven by the major transcription factors ETS1 (patient clusters 1, 2), ETS2 (all patient clusters), TP53 (patient clusters 1, 2), STAT3 (patient clusters 2, 3), SMAD3 (all patient clusters), CREB1 (patient clusters 1, 2, 3, 4), CREM (patient cluster 1, 2, 3, 4), and IZKF1 (all patient clusters) (Fig. 3E,F). These results indicate a spectrum of gene regulatory interactions impacted by non-coding SNPs that define distinct CD patient clusters. Comparing patient-specific differences and similarities between these gene regulatory networks could provide a better understanding of how patient-specific combinations of non-coding SNPs contribute to heterogeneous pathomechanisms in CD. Similar insights have been observed in oncology, where gene regulatory networks stratified patients with various types of cancer into distinct subgroups with prognostic implications (Nakazawa et al, 2021).

## Patient-specific clusters of SNP-propagated gene regulatory networks correspond to cell type-specific gene dysregulation patterns in CD patients

We posited that the identified CD patient clusters of SNP-propagated gene regulatory networks may have cell type-specific

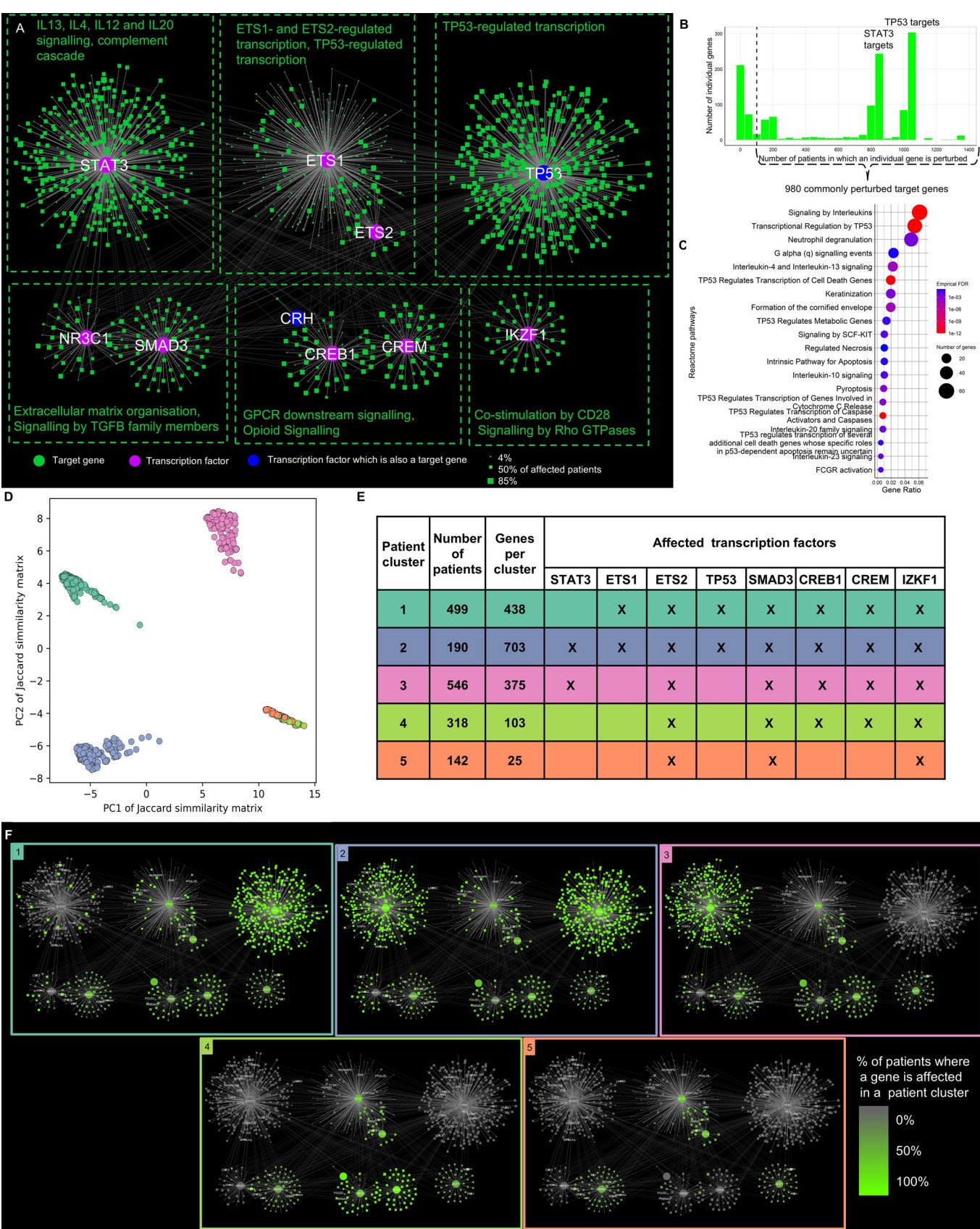

◀ **Figure 3.  SNP-propagated gene regulatory network divides patients into five clusters in Crohn's disease.**

(**A**) The SNP-propagated gene regulatory network in Crohn's disease (CD). Pink nodes represent transcription factors (TFs), while green nodes are target genes of the TFs, and blue nodes are both. The size of target gene nodes corresponds to the number of patients having the SNP-propagated target genes. The network represents patient-specific gene regulatory modules. (**B**) Frequency distribution of SNP-propagated TF-target genes in the CD patient cohort. (**C**) The enriched Reactome pathways of the commonly perturbed SNP-propagated TF-target genes in the CD patient cohort. (**D**) Principal component analysis of the Jaccard similarity matrix between the patient-specific gene regulatory networks. Each point represents a patient, and the colours are according to k-means clustering, where k is 5. (**E**) Patient-specific clusters and their cluster-representative SNP-propagated TFs. (**F**) Cluster-representative gene regulatory networks underpinning each patient cluster. Source data are available online for this figure.

effects. To test this hypothesis, we leveraged two large publicly available single-cell transcriptomics datasets (Kong et al, 2023; Krzak et al, 2023) comprising CD patients ($n = 46$ and $n = 26$, respectively) and healthy controls ($n = 25$ and $n = 25$, respectively) to determine if the target genes from each patient-specific cluster of SNP-propagated regulatory networks overlapped with CD single-cell differentially expressed genes (DEGs). In the Kong et al datasets, patient-specific clusters mapped to DEGs in a cell type- and location-specific manner (Fig. 4; Dataset EV4).

In the Kong et al cohort, target genes from CD patient clusters 2, 3, and 4 were overrepresented in DEGs from both colonic and ileal cell types, whereas cluster 5 mapped almost exclusively to colonic cell types. By contrast, cluster 1 target genes did not show enrichment in any of the cell types evaluated. In the colon, macrophages, mast cells, and endothelial cells were most frequently enriched across patient clusters, whilst in the ileum, Paneth cells were the most commonly enriched cell type. Notably, these cell types were particularly strongly enriched alongside colonic inflammatory fibroblasts and LGR5+ OLFM4+ stem cells in patient clusters 2 and 3. Although clinical metadata reported in these studies were limited, these location-specific enrichments may reflect ileal versus colonic CD phenotypes, consistent with large-scale genotype–phenotype studies showing that ileal and colonic CD represent genetically distinct subtypes (Cleynen et al, 2016). In the independent Krzak et al CD cohort (Appendix Fig. S3A), two clusters (clusters 2 and 4), both regulated by STAT3, were overrepresented in B cells, goblet cells, and CD4+ CXCR6+ memory T cells.

We further validated cluster-representative TFs by inferring TF activity from the gene expression data of CD patients from the Kong et al and Krzak et al studies (Appendix Figs. S3B and S4). Strikingly, we found that out of the eight major cluster-representative transcription factors (ETS1, ETS2, STAT3, IZKF1, SMAD3, CREB1, CREM, and P53), which had more than five target genes, all were predicted to have perturbed activities in the Kong et al dataset. In the Krzak et al dataset, seven out of the eight TFs were predicted to have altered activities. Again, there were notable differences in TF activities between ileal and colonic tissues. STAT3, a major downstream TF in cytokine signalling (Hillmer et al, 2016), was predicted to be dysregulated in almost all cell types in ileal CD versus control tissues from the Krzak et al dataset. Similarly, in the Kong et al dataset, STAT3 perturbation was observed in ileal immune cells but not in colonic tissues (Appendix Fig. S4). Meanwhile, SMAD3 had a strong stromal signal, particularly in myofibroblasts, in both datasets. In summary, these results show that signals from CD-associated non-coding SNPs propagate towards downstream gene regulatory networks through certain TFs in a cell type- and location-specific manner.

## Non-coding SNPs in UC impact signalling processes involving chemokine signalling, apoptosis, microbial responses, and stromal processes

We next investigated how non-coding SNPs may impact signalling processes in UC patients. We identified 38 non-coding SNP-affected proteins in our cohort of 941 UC patients (Dataset EV1), which we used as seeds for the heat diffusion algorithm to reconstruct a SNP-propagated signalling network containing 118 proteins and 246 protein–protein interactions. This UC SNP-propagated signalling network comprised four major functional modules (Fig. 5A) relating to (i) chemokine receptor and GPCR signalling; (ii) apoptosis and cell death pathways; (iii) HOX gene activation; and (iv) TLR cascade. A minor module related to macroautophagy was also observed (for other modules, see Dataset EV2). Most of these pathways are known to be involved in UC pathogenesis (Kobayashi et al, 2020), including macro-autophagy, for which there is increasing evidence of dysregulation in UC (Shao et al, 2021) following its well-established role in CD (Larabi et al, 2020).

In total, 95 proteins were predicted to be perturbed in at least 100 UC patients in our cohort (Fig. 5B). These commonly perturbed proteins in UC were functionally enriched for pathways related to microbial responses (parasitic infection, TLR9 cascade), stromal processes (platelet aggregation/activation, VEGF signal-ling), and macroautophagy (Fig. 5C; Dataset EV3). As these are highly relevant pathogenic pathways in UC (Sánchez-Muñoz et al, 2010; Tolstanova et al, 2010; Ungaro et al, 2017), the systems genomics approach presented here provides novel insights into the molecular mechanisms linking non-coding UC-associated risk variants with known pathogenic drivers of the disease. These SNP-propagated proteins were significantly enriched for the transcriptomics signature associated with inflamed UC samples but not non-inflamed samples, suggesting that the SNP-propagated signalling network is more relevant in UC pathogenesis in the context of active rather than quiescent disease (Appendix Table S2).

## Patient-specific SNP-propagated gene regulatory networks stratify UC patients into eight clusters

We next evaluated the downstream gene regulatory layer in UC. The UC SNP-propagated gene regulatory network consisted of 2813 genes and 3895 TF-TG interactions. The network was formed by 12 TFs, 2795 target genes, and 6 genes which functioned as both. Five gene regulatory modules were identified in this regulatory network (Fig. 6A). These modules corresponded to various biological functions: (i) cytokine signalling and TLR cascade (regulated by NFκB1, EGR1, DNMT3B, and other TFs); (ii)

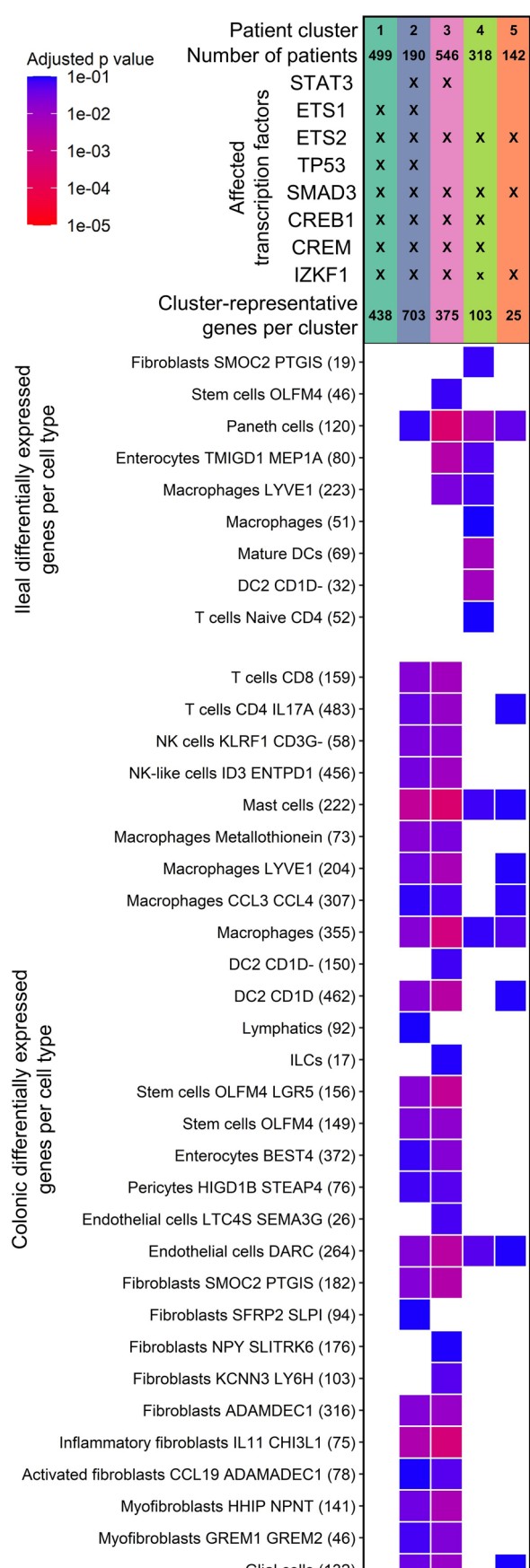

**Figure 4. Patient-specific clusters correspond to cell type-specific differentially expressed genes in Crohn's disease.**

The x axis corresponds to the various patient-specific clusters based on the SNP-propagated gene regulatory networks. The y axis denotes intestinal cell types with differentially expressed genes (DEGs) from Crohn's disease (CD) patients. DEGs were defined by comparing the expression of healthy versus non-inflamed CD biopsies from the Kong et al study (|log2FC| > 0.5, Benjamini–Hochberg corrected Wilcoxon rank-sum test P < 0.1 (Kong et al, 2023)). The number of DEGs between healthy and non-inflamed CD cells in a given cell type is shown in brackets. If a square has a colour, then the cluster-representative genes are overrepresented in the cell type-specific differentially expressed gene set (Benjamini–Hochberg adjusted hypergeometric test P < 0.1). If a square is black, that means a cluster is not overrepresented in the cell type-specific CD signature. FC fold change. Source data are available online for this figure.

activation of HOX genes (regulated by CEBPB and retinoic acid receptor (RAR) family TFs); (iii) processing of the mitochondrial calcium uniporter complex, SMDT1 (regulated by E2F1); and (iv) xenobiotics and synthesis of arachidonic acid derivatives (regulated by ETS1, ETS2, AHR, and NRF2). A fifth module was regulated by FOXP3, but its target genes were not significantly enriched with any pathways (Dataset EV2). Thus, as seen in CD, the SNP-propagated regulatory network analysis revealed that non-coding SNPs perturb different but highly relevant pathogenic pathways at the gene regulatory layer compared with the signalling layer in UC.

Unlike the UC SNP-propagated signalling network, the frequency distribution of target genes in the SNP-propagated regulatory network had a bimodal distribution (Fig. 6B)—one peak with fewer than 50 patients, and the other peak involving 669 patients. The latter peak comprised the target genes of FOXP3. Another smaller peak involving 555 patients corresponded to NFκB1 target genes. We next identified target genes that were perturbed in at least 100 UC patients in our cohort. Similar to the UC SNP-propagated signalling network, these 2244 commonly perturbed target genes were enriched for the transcriptomics signature of inflamed UC intestinal tissues but not non-inflamed samples (Appendix Table S2). Functional annotation of these target genes revealed that they were involved in IL23 and IL10 signalling (Fig. 6C; Dataset EV3), indicating that in a substantial proportion of UC patients, the underlying genetic background may contribute to the dysregulation of critical pro- and anti-inflammatory pathways.

Similar to our observations in CD, clustering of SNP-propagated gene regulatory networks stratified UC patients more effectively than SNP-propagated signalling networks (Appendix Fig. S2B), yielding eight distinct patient clusters (Fig. 6D). This stratification was based on the combination of involved gene regulatory modules specifically driven by FOXP3, ETS2, NFκB1, CEBP, and DNMT3B (Fig. 6E,F). Unlike CD, only three patient clusters (clusters 1, 4, and 8) corresponded with UC cell type-specific signatures obtained from the Smillie et al single-cell dataset (Appendix Fig. S5) (Smillie et al, 2019), likely due to low statistical power of the Smillie et al dataset (n = 17 UC patients) compared to the Kong et al dataset (n = 46 CD patients).

## Patient-specific clusters of SNP-propagated gene regulatory networks correspond to therapeutic response to ustekinumab therapy in UC patients

To understand whether the stratified UC patient clusters correspond to clinically relevant phenotypes, we investigated their

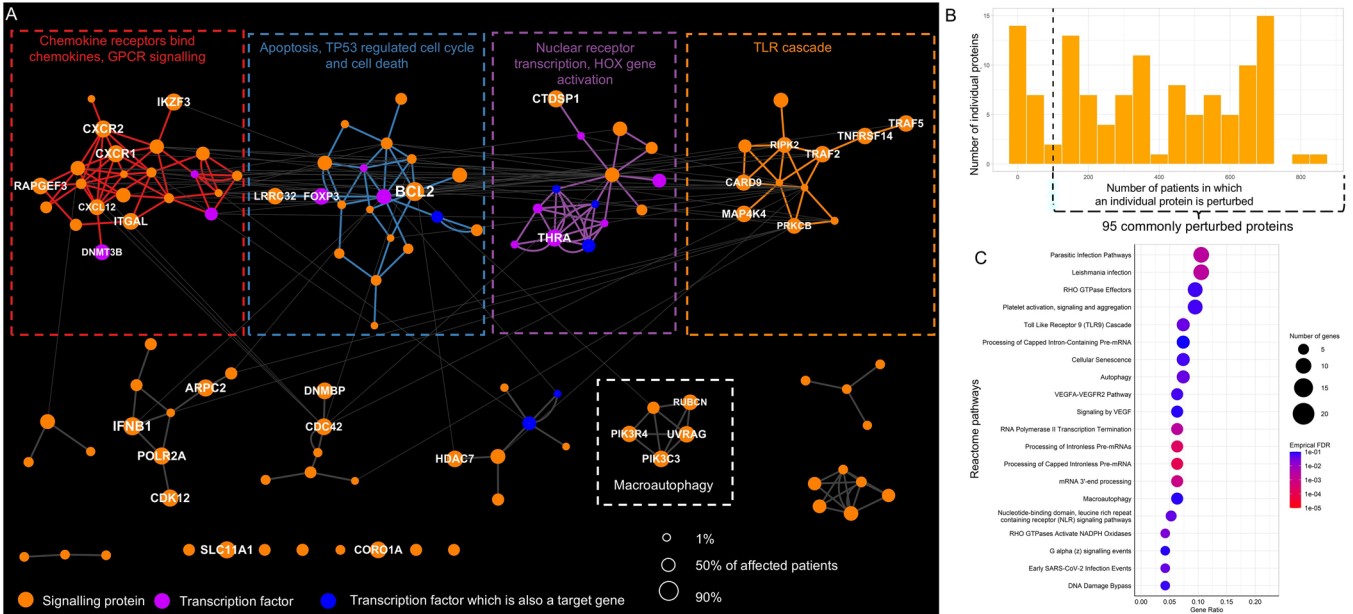

**Figure 5. Ulcerative colitis-associated non-coding SNPs affect T-cell activation, NFκB signalling, cytokine signalling, and chemotaxis.**

(A) SNP-propagated signalling network in ulcerative colitis (UC) proteins grouped by Girvan–Newman modularisation. The largest four modules functionally annotated using Reactome enrichment, and their edges are coloured. Node size corresponds to the number of patients harbouring the SNP-propagated protein. A macroautophagy-related module has been annotated separately. (B) Frequency distribution of SNP-propagated proteins in the UC patient cohort. (C) The enriched Reactome pathways of the most commonly affected SNP-propagated proteins in the UC patient cohort. Source data are available online for this figure.

potential influence on therapeutic response. We analysed data from the randomised placebo-controlled UNIFI trial, which investigated the efficacy and safety of ustekinumab (anti-IL12/IL23 p40 antibody) in moderate-to-severe UC patients (Sands et al, 2019). Clusters 4, 6, 7, and 8 in our cohort were significantly enriched for DEGs distinguishing responders from non-responders in the UNIFI trial (Benjamini–Hochberg corrected hypergeometric test $P < 0.05$, Fig. 6E,F). This suggests that patients in these clusters may have a differential response to ustekinumab therapy, as their genetic background perturbs the very genes that differentiate between treatment response and non-response. Notably, these clusters were all characterised by gene regulatory modules regulated by NFκB1, whereas no such enrichment was observed in the other clusters, pointing to NFκB1-driven regulation as a potential determinant of ustekinumab response. Whilst the precise mechanisms require further experimental evaluation, these findings suggest a hitherto unrecognised genetic basis for therapeutic response to ustekinumab in UC patients, which could be leveraged to inform future precision medicine strategies.

## CD and UC non-coding SNPs converge on the same signalling pathways, but they perturb distinct gene regulatory modules

After separately analysing the downstream effect of non-coding SNPs from CD and UC patients, we compared the two IBD cohorts with each other. A comparison of the two cohorts revealed only six non-coding SNPs shared between our CD and UC patients (Appendix Fig. S1), highlighting the different genetic predispositions underlying the two diseases. Then, to compare CD and UC on

a more systemic level, we generated a combined IBD SNP-propagated signalling network and a combined IBD SNP-propagated gene regulatory network.

The combined IBD SNP-propagated signalling network revealed four major modules of perturbed signalling proteins that had shared nodes and edges between both diseases. These modules functionally corresponded to pathways relating to (i) interleukin signalling and apoptosis; (ii) TLR and NFκB signalling; (iii) GPCR signalling; and (iv) Wnt signalling (negative regulation of TCF-dependent signalling by DVL-interacting proteins) and parasitic infection (Fig. 7A; Dataset EV2). The common proteins in the combined IBD SNP-propagated signalling network included those known to be involved in IBD pathogenesis, such as JAK2, JAK3, BCL2, MAPK8, and KIT, although different signalling interactions connected CD and UC non-coding SNPs to these proteins. These shared proteins functionally annotated for interleukin signalling (in particular IL17 signalling), TLR4 cascade, and MAPK activation (Fig. 7B). The combined SNP-propagated signalling network also contained small, disease-specific network modules such as the retinoic acid signalling module in UC or the base excision repair module in CD, suggesting that these processes are likely to have disease-specific genetic associations from multiple SNPs. Interestingly, modules associated with autophagy were found in both UC ("macroautophagy" underpinned by autophagy proteins such as RUBCN, UVRAG, and BECN1) and CD ("gap junction degradation" underpinned by the key autophagy protein ATG16L1).

Unlike the SNP-propagated signalling network, the combined IBD SNP-propagated gene regulatory network revealed clear differences between CD and UC. This was because signals propagated by non-coding SNPs perturbed downstream TFs which

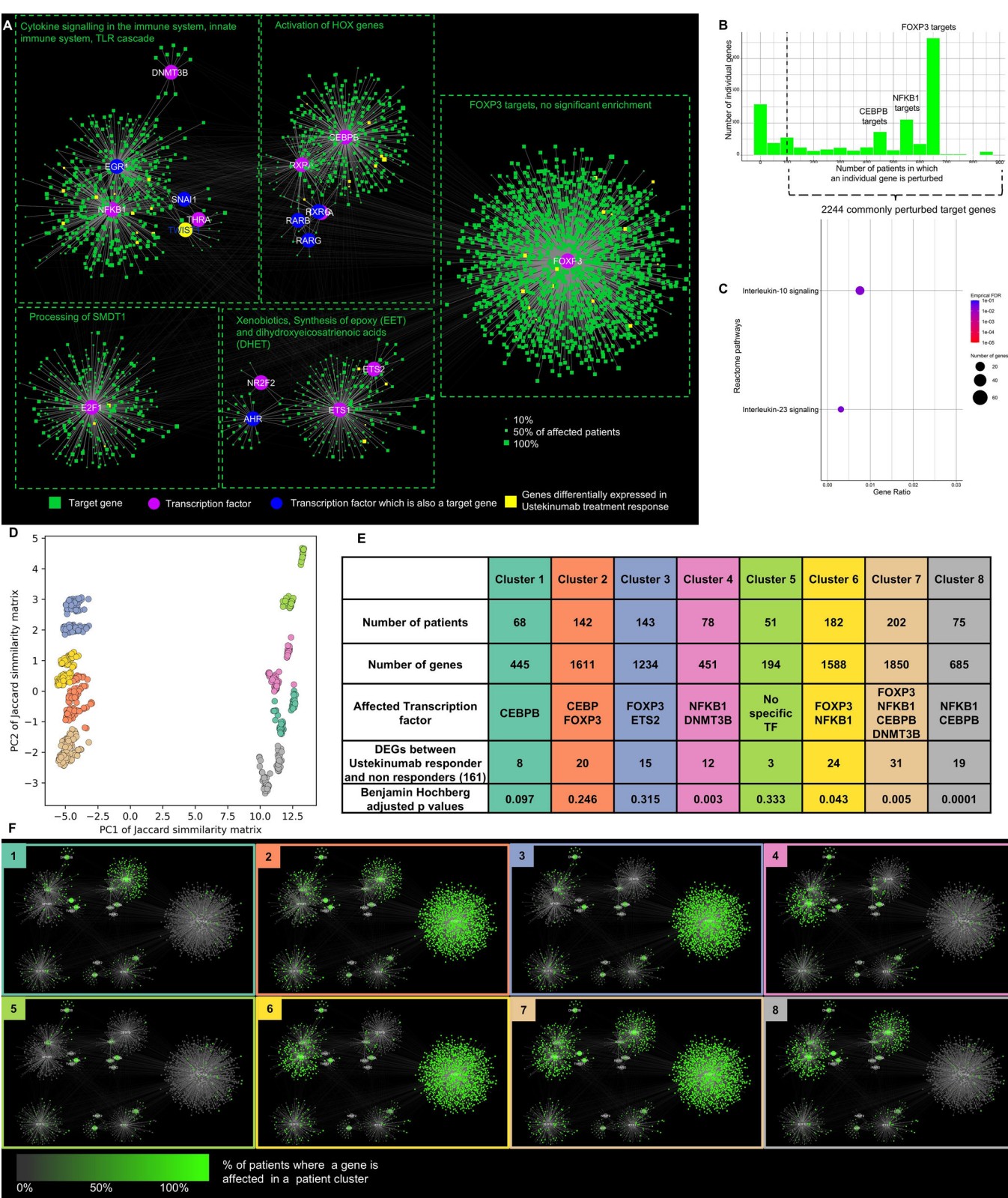

◄ **Figure 6.    SNP-propagated gene regulatory network divides ulcerative colitis patients into eight clusters based on three main transcriptional regulators: CEBP, NFκB1 and FOXP3.**

(A) The SNP-propagated gene regulatory network in ulcerative colitis (UC) patients. Pink nodes represent transcription factors (TFs), while green nodes are TF target genes, and blue nodes are both. The size of target gene nodes represents the number of patients having the SNP-propagated target genes. (B) Frequency distribution of SNP-propagated TF-target genes in the UC patient cohort. (C) Significantly enriched Reactome pathways in the most commonly perturbed SNP-propagated target genes in the UC patient cohort. (D) Principal component analysis of the Jaccard similarity matrix between the patient-specific gene regulatory networks. (E) Patient-specific clusters and their cluster-representative TFs. The enrichment of differentially expressed genes comparing ustekinumab response versus non-response is also shown here. (F) Cluster-representative affected nodes. The colour corresponds to how often each gene is affected in a given patient cluster. Source data are available online for this figure.

had minimal overlap between CD and UC, leading to largely distinct downstream gene regulatory interactions (Fig. 7C; Appendix Fig. S6). STAT3 and TP53 were identified as TFs specifically perturbed in CD, whilst CEBPB, FOXP3, and NFκB1 were TFs specifically perturbed in UC. The only overlap observed was in ETS1- and ETS2-regulated target genes, which were perturbed in both CD and UC SNP-propagated gene regulatory networks (Appendix Fig. S6).

In summary, our comparative analysis revealed that while UC and CD non-coding SNPs converge on multiple common signalling pathways (albeit through different network routes), they predominantly perturb distinct gene regulatory modules.

### Sensitivity and robustness analyses of network propagation results

As with any network-based analysis, it is important to assess whether source resources or publication bias may have influenced the results. A previous, independent benchmarking study reported that OmniPath had one of the lowest correlations between the degree of protein nodes and publication counts associated with proteins in comparison to four other signalling network resources, indicating that it is less influenced by publication bias (Arici and Tuncbag, 2021). To systematically evaluate the sensitivity of our findings to the networks we selected (i.e. OmniPath and DoRoThEA), we repeated the iSNP network propagation using two alternative signalling networks (i.e. Reactome (Milacic et al, 2024) and STRING (Szklarczyk et al, 2023)) and one alternative gene regulatory network (i.e. CollecTRI (Müller-Dott et al, 2023)).

While STRING produced the largest signalling network, OmniPath provided the highest representation of SNP-affected proteins in both UC (51 out of 95 possible SNP-affected proteins) and CD (30 out of 47 SNP-affected proteins) (Appendix Fig. S1), indicating that OmniPath had the most specific coverage of SNP-affected proteins among the evaluated signalling networks. In UC, OmniPath in combination with DoRoThEA, yielded the greatest number of perturbed target genes in the SNP-propagated gene regulatory network. In CD, Reactome and STRING produced a higher number of perturbed proteins in the SNP-propagated signalling network. In both UC and CD, DoRoThEA identified more perturbed target genes in the SNP-propagated gene regulatory network than CollecTRI. There were varying degrees of overlap in proteins and TF-TGs shared between the databases (Appendix Fig. S8), indicating that the choice of network resources used to construct the SNP-propagated networks can influence propagation outcomes. We clustered the patients based on various signalling and regulatory network pairs as well, showing that gene regulatory networks produced better clustering compared to

signalling networks in every network combination (Appendix Fig. S9A–D).

In addition, SNP-propagated networks were constructed using degree-matched random seeds to test the robustness of our findings to network topology (Appendix Fig. S9E–H). These analyses identified more perturbed proteins and TF-TGs than the non-degree-matched control networks. The non-degree-matched network was a subset of the degree-matched networks, irrespective of the underlying network resources used (Appendix Fig. S10). The patient clustering results were similar in degree-matched networks compared to non-degree-matched networks (Appendix Fig. S11). Overall, these findings indicate that the identified perturbed nodes and patient clusters remain largely robust to variations in network topology.

## Discussion

In this study, we present a novel systems genomics workflow that addresses key challenges in complex disease genetics, including missing regulation and polygenicity, to disentangle the putative biological mechanisms linking non-coding genetic variants to pathogenic pathways. Previous approaches have largely focused on integrating GWAS signals with chromatin accessibility, expression quantitative trait loci (eQTLs), and gene expression data (Cano-Gamez and Trynka, 2020), or on mapping SNP-affected genes to PPI networks (Kara et al, 2019; Cowen et al, 2017; Huang et al, 2018) to identify disease-relevant genes or pathways. However, these strategies typically overlook the downstream impact of genetic risk loci across both signalling and gene regulatory layers, which is critical for understanding how non-coding genetic risk variants ultimately reshape cellular pathways and contribute to disease phenotypes. By leveraging network propagation across both layers, our workflow models the holistic impact of disease-associated non-coding SNPs, extending beyond local interacting neighbours to encompass the global topology of cellular signalling and gene regulatory networks.

Using this methodology, we performed the largest patient-specific systems genomics analysis in chronic IMIDs to date, analysing genotype data from over 2500 individuals with the two major forms of IBD: CD and UC (de Lange et al, 2017; Huang et al, 2017). Despite the success of GWAS in identifying more than 240 susceptibility variants in IBD (de Lange et al, 2017), genotype–phenotype associations have been disappointingly sparse, both in terms of mechanistic insights and clinical outcomes (Cleynen et al, 2016). Similar challenges have also been observed across other chronic IMIDs, including rheumatoid arthritis, multiple sclerosis, and type 1 diabetes (Stankey and Lee, 2023).

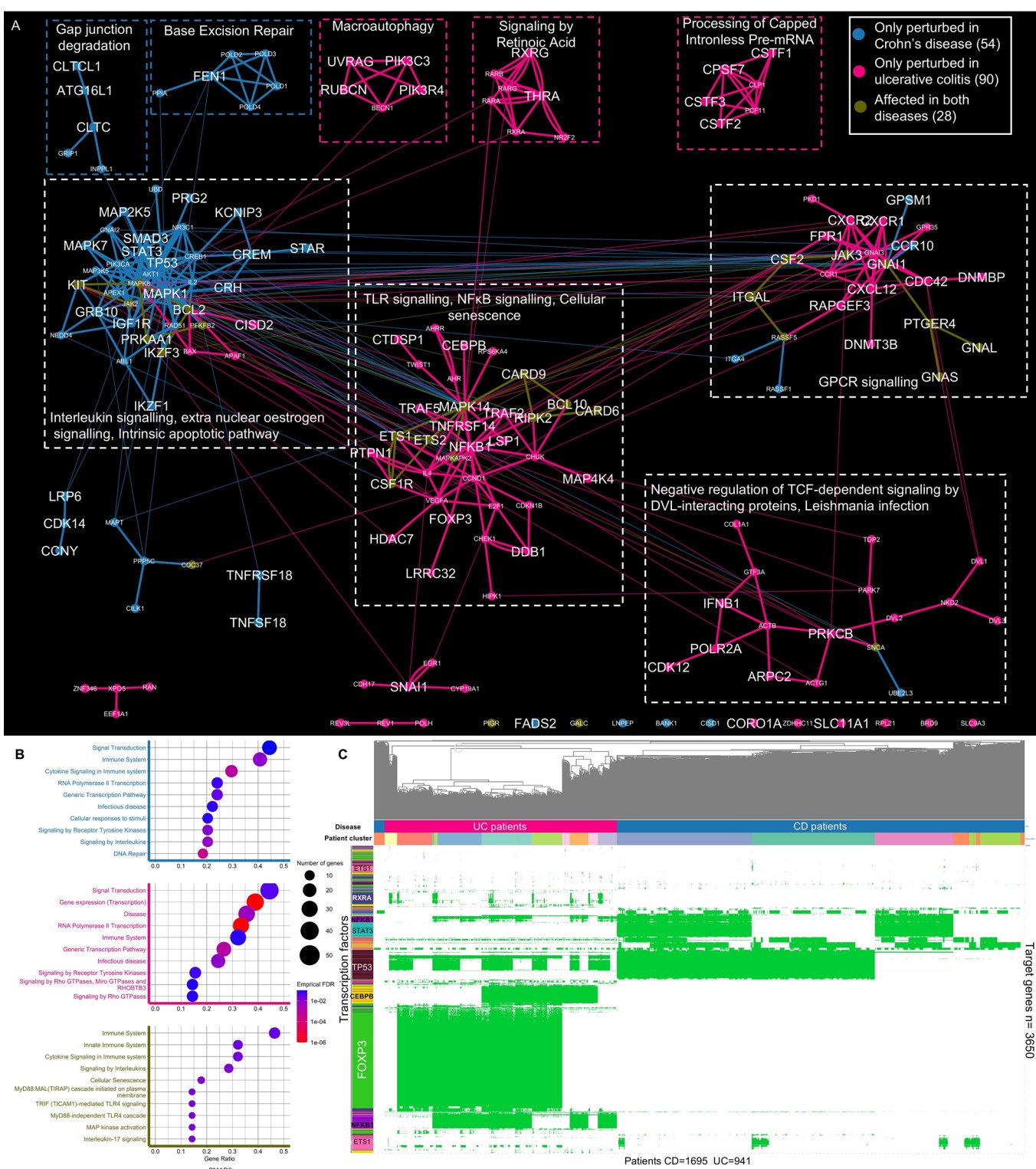

As a result, the initial optimism surrounding the use of genomics for identifying patient-specific pathomechanisms and advancing precision medicine efforts in chronic IMIDs has been dampened, prompting many in the field to shift focus towards other omic layers (Noor et al, 2022). In this work, however, we demonstrate that patient genotype can offer crucial biological and translational insights when analysed through the lens of systems genomics. By unravelling the heterogeneous signalling and gene regulatory mechanisms connecting individual patient genotypes with phenotypes in the prototypical chronic IMIDs of UC and CD, our workflow highlights the potential of genotype-based stratification to inform future personalised medicine approaches. This is urgently

◄ **Figure 7. Crohn's disease and ulcerative colitis patients share several pathways in their SNP-propagated signalling networks, whilst their SNP-propagated gene regulatory networks reveal distinct transcription factor-target gene interactions.**

(**A**) Combined SNP-propagated signalling network of Crohn's disease (CD) and ulcerative colitis (UC) patients. Node size corresponds to the number of CD or UC patients having the SNP-propagated protein. Modules shared between UC and CD are shown with a white border, whilst disease-specific modules are depicted with a pink border (UC) and a blue border (CD). (**B**) Reactome pathway enrichment of the SNP-propagated proteins perturbed in both diseases. (**C**) Heatmap of SNP-propagated TFs and their target genes in CD and UC patients. The columns represent patients clustered according to the TF-target genes present in the SNP-propagated gene regulatory network in CD (blue) and UC (purple). Rows represent SNP-propagated TFs. Within the heatmap, green colour indicates TF-target gene interactions present within the SNP-propagated gene regulatory network, whilst white indicates interactions that are not present. Source data are available online for this figure.

required given the significant therapeutic ceiling effect currently observed across multiple chronic IMIDs (Guillo et al, 2023; Raine and Danese, 2022; Smolen et al, 2018).

The presented analysis revealed that UC- and CD-associated non-coding genetic variants propagate towards and perturb key pathogenic pathways known to be implicated in IBD pathogenesis. At the signalling layer, CD-associated non-coding SNPs were predicted to impact proteins involved in pathways known to drive CD pathogenesis such as interleukin signalling (including IL2), DNA repair, and autophagy (Lassen et al, 2014; Tschurtschenthaler et al, 2017; Lee et al, 2025) (Fig. 2). Interestingly, while autophagy dysfunction in CD is traditionally attributed to the *ATG16L1* coding risk variant (Hampe et al, 2007), our findings suggest that autophagy signalling may also be disrupted in patients carrying non-coding SNPs (e.g. rs3792112 and rs3828309). In UC, the propagation of non-coding SNPs to the signalling layer also emphasised pathways linked to UC pathogenesis, including microbial responses (TLR4 and TLR9 signalling (Levin and Shibolet, 2008)), chemokine and GPCR signalling, and stromal processes such as VEGF signalling (Tolstanova et al, 2010) (Fig. 5). Combined analysis and visualisation of both UC and CD SNP-propagated signalling networks revealed that distinct non-coding SNPs associated with UC and CD converge on many of the same cellular signalling pathways such as interleukin (particularly IL17), TLR, NFκB, JAK, MAPK and WNT signalling, but through different disease-specific pathomechanistic routes (Fig. 7A). These findings also align with the clinical efficacy of therapies targeting these convergent pathways in both UC and CD, such as JAK inhibitors (e.g. upadacitinib) that directly inhibit JAK signalling, and biologics targeting the IL23 axis (e.g. ustekinumab or risankizumab) that indirectly modulate IL17-driven inflammatory signalling (Noviello et al, 2021; Bourgonje et al, 2025).

At the downstream gene regulatory layer, non-coding SNPs were predicted to impact several cellular processes implicated in IBD pathogenesis, driven by largely distinct transcriptional regulators in CD and UC. For instance, in CD, regulatory networks were dominated by hub TFs such as STAT3 (regulating interleukin signalling and neutrophil degranulation), SMAD3 (regulating ECM organisation), and p53 (regulating cell death pathways), whilst in UC, NFκB1 and EGR1 (regulating cytokine signalling and TLR cascade), ETS1 (regulating xenobiotics and fatty acid synthesis), and FOXP3 emerged as some of the key perturbed transcriptional regulators. Interestingly, ETS2, recently identified as a master regulator in CD (Stankey et al, 2024), and ETS1 were the only TFs predicted to be perturbed by non-coding SNPs in both diseases. The near-complete non-overlap of TF-driven modules between CD and UC highlights the regulatory specificity of each disease, which

may underpin their distinct immunopathology and clinical phenotypes despite shared signalling convergence.

Consistent with these observations, SNP-propagated gene regulatory networks stratified our patient cohort more effectively than either SNP-propagated signalling networks or neighbouring SNP-affected genes (Appendix Fig. S7), revealing heterogeneous gene regulatory pathomechanisms across patient subgroups. In CD, we contextualised these patient clusters to specific cell types and spatial locations along the gastrointestinal tract using single-cell transcriptomics data (Fig. 4; Appendix Figs. S3 and S4). In the colon, macrophages, mast cells, and endothelial cells were most frequently enriched for cluster-representative SNP-propagated gene regulatory networks, whereas in the ileum, Paneth cells were most commonly enriched. These findings suggest that distinct cell types across different regions of the gastrointestinal tract may exhibit perturbed functions depending on the individual patient's specific genetic background. Moreover, in UC, we demonstrated that SNP-propagated gene regulatory networks overlapped with therapeutic response signatures to ustekinumab (anti-IL12/23 p40 monoclonal antibody), which is a widely used biologic in current clinical practice. Notably, only UC patient clusters with NFκB1-perturbed modules in their SNP-propagated gene regulatory networks were enriched for genes differentiating ustekinumab responders from non-responders in the UNIFI trial (Sands et al, 2019) (Fig. 6E,F), suggesting a possible mechanistic link between NFκB1 regulation and ustekinumab efficacy. If validated in independent cohorts, such findings could support genetically informed patient stratification for advanced therapies in UC, addressing a pressing need in IBD precision medicine.

The presented two-step propagation approach allowed us to capture SNP effects separately at the downstream signalling and gene regulatory layers, yielding complementary and biologically meaningful insights into UC and CD pathogenesis that might be obscured in single-layer or combined network analyses. To our knowledge, direct benchmarking of two-step versus combined network propagation has not yet been reported in complex diseases. However, related work outside of this context supports the value of multilayered approaches. For example, TieDIE (Paull et al, 2013) applied separate diffusion processes from signalling inputs and regulatory outputs (such as TFs), achieving higher precision in cancer datasets compared with single diffusion on combined networks. Importantly, TieDIE demonstrated that separating diffusion reduced bias toward highly connected nodes, which in a combined network tend to dominate the propagation and mask disease-specific mechanisms. Similarly, random walks on multiplex biological networks have been found to outperform those on monoplex or aggregated networks, as they preserve the complementary contributions of each interaction type without losing

information from collapsing layers (Valdeolivas et al, 2019). Collectively, these results indicate that multilayer propagation captures specific links between genetic alterations and their transcriptional consequences, which may be missed in single-layer or aggregated network analyses.

Whilst our systems genomics analysis led to plausible predictions for the impact of non-coding SNPs in IBD, we acknowledge the limitations of the presented study. First, we did not integrate non-regulatory SNPs such as coding risk-variants or fully account for epigenetic architecture beyond TF binding (such as chromatin accessibility or conformation), which could further refine our model. Second, the lack of well-powered UC single-cell transcriptomics datasets restricted our ability to make cell type-specific predictions for the various UC patient clusters identified in our analysis, though validation against bulk transcriptomics data supported disease relevance. Third, the limited availability of clinical metadata in our patient cohort prevented correlation of patient-specific regulatory clusters with clinical features such as disease activity or disease extent. Emerging large IBD patient biobanks, such as the IBD BioResource (Parkes and IBD BioResource Investigators, 2019), which capture both genetic information and detailed clinical metadata, will enable future validation of genotype-perturbed signalling and gene regulatory mechanisms and their association with IBD clinical phenotypes. Fourth, our equivalent treatment of heterozygous and homozygous non-coding variants does not account for allele dosage effects, but given the limited quantitative understanding of dosage effects on TF binding, we adopted this conservative approach. Fifth, although the inclusion of fine-mapped IBD-associated non-coding variants provided a high-confidence variant set to model potential downstream signalling and regulatory consequences, false positives or negatives remain possible as fine-mapping cannot definitively prove causality. Resolving the functional effects of SNPs at individual loci on TF motifs or miRNA binding was beyond the scope of this work but remains an important next step. In addition, while our choice of network resources (i.e., OmniPath and DoRoThEA) maximised coverage of SNP-affected proteins and perturbed TF-TGs, and enabled comparability across UC and CD, propagation outcomes varied across databases, highlighting how resource selection can influence results (Appendix Figs. S1, S8, and S9 and see "Methods"). We also found that degree-matched random seeds propagated signals more broadly but recovered most of the perturbed nodes identified in the non-degree-matched analyses (Appendix Fig. S10). Finally, while one-step heat propagation at the gene regulatory layer better reflected the more promiscuous interactions that are known to occur between TFs and their targets, we acknowledge that global propagation methods which leverage the full gene regulatory network topology could, in the future, provide a complementary method to prioritise targets of pleiotropic TFs. Taken together, these considerations highlight the methodological and data-driven boundaries of the current study, while also outlining clear opportunities for refinement as larger clinically annotated genomic datasets become available.

Our findings also open up several avenues for experimental follow-up that could proceed along complementary axes. First, multi-omics validation in independent IBD patient cohorts would allow iSNP to identify and validate the functions of cluster-representative TFs by integrating genomic data with transcriptomics, which provides indirect measures of TF activity, and epigenetic data such as ChIP-seq, which confirms TF binding in relevant cell types. Second, in vitro functional perturbation using CRISPR/Cas9-mediated knockdown or overexpression of predicted TFs (e.g., STAT3, FOXP3) in patient-derived intestinal organoids or primary immune cells could assess downstream consequences, complemented by massively parallel reporter assays (MPRA) to functionally test non-coding SNPs and CUT&RUN to map TF occupancy at predicted regulatory sites (Stankey and Lee, 2023; Lee et al, 2024). Encouragingly, the field is already moving in this direction, exemplified by a recent landmark study mapping the pleiotropic intergenic risk variant at chr21q22 to ETS2 dysfunction in inflammatory macrophages in CD (Lee et al, 2024). Third, in vivo testing, using murine colitis models (e.g., DSS, TNBS, or genetically modified strains (Kiesler et al, 2015)) and complementary zebrafish IBD reporter lines (Jijon et al, 2018) harbouring human TFBS regions with or without IBD-associated SNPs could be harnessed to interrogate the role of genetic variants in intestinal inflammation. The workflow developed in this study is openly available (https://github.com/korcsmarosgroup/iSNP) and could accelerate such efforts.

In summary, by embedding and propagating signals from non-coding SNPs across downstream cellular signalling and gene regulatory networks, our integrated network analysis provided plausible mechanisms connecting disease-associated genotype with pathogenic pathways. These mechanisms were patient- and/or cell type-specific, and also associated with clinically relevant phenotypes in both UC and CD. This work lays the foundation for addressing the challenges of missing heritability (Manolio et al, 2009; Boyle et al, 2017), missing regulation (Connally et al, 2022), and polygenicity, which together pose a major barrier for the translation of genomics into precision medicine strategies in chronic IMIDs. The presented study and workflow offer an important advancement towards this goal and can be readily applied to unravel the genetic underpinnings of other complex diseases.

# Methods

**Reagents and tools table**

| Reagent/resource | Reference or source | Identifier or catalogue number |
| --- | --- | --- |
| **Experimental models** | | |
| Nil | | |
| **Recombinant DNA** | | |
| Nil | | |
| **Antibodies** | | |
| Nil | | |
| **Oligonucleotides and other sequence-based reagents** | | |
| Nil | | |
| **Chemicals, enzymes and other reagents** | | |
| Nil | | |
| **Software** | | |
| Cytoscape version 3.10.1 | Shannon et al, 2003 | |

| Reagent/resource | Reference or source | Identifier or catalogue number |
|---|---|---|
| Python version 3.11.5 | https://www.python.org/doc/versions/; https://docs.python.org/3/whatsnew/3.8.html | |
| R version 4.2.2 | R Core Team, 2015 | |
| HEDD database (date of download: 13/08/2023) | Wang et al, 2018 | |
| miRanda version 1.9 | Hsu et al, 2006 | |
| JASPAR database (date of download: 10/10/2023) | Castro-Mondragon et al, 2022 | |
| Regulatory Sequence Analysis Tools version 2018.8.1 | Medina-Rivera et al, 2015 | |
| Find Individual Motif Occurrences tool version 5.1.1 | Grant et al, 2011 | |
| OmniPath database (date of download: 27/10/2021) | Türei et al, 2021 | |
| DoRothEA database (date of download: 27/10/2021) | Garcia-Alonso et al, 2019 | |
| clusterProfiler version 4.6.0 | Wu et al, 2021 | |
| enrichplot version 1.18.3 | https://bioconductor.org/packages/release/bioc/html/enrichplot.html | |
| org.Hs.eg.db version 3.16.0 | https://bioconductor.org/packages/release/data/annotation/html/org.Hs.eg.db.html | |
| AnnotationDBI version 1.60.0 | https://bioconductor.org/packages/release/bioc/html/AnnotationDbi.html | |
| clusterMaker version 2.3.4 | Morris et al, 2011 | |
| Scipy pdist version 1.14.1 | Virtanen et al, 2020 | |
| scikit-learn version 1.5.1 | Pedregosa et al, 2011 | |
| Seurat version 4.3.1 | Hao et al, 2021 | |
| Python DEseq2 version 0.4.10 | Muzellec et al, 2023 | |
| R Deseq2 version 1.42.1 | Love et al, 2014 | |
| ENSEMBL Biomart package version 2.58.2 | Smedley et al, 2009 | |
| R DecoupleR version 2.9.7 | Badia-I-Mompel et al, 2022 | |
| decoupler-py version 1.7.0 | Badia-I-Mompel et al, 2022 | |
| pybiomart package version 0.2.0 | https://jrderuiter.github.io/pybiomart/ | |
| GEO2R | Barrett et al, 2013 | |
| limma version 3.46.0 | Ritchie et al, 2015 | |

| Reagent/resource | Reference or source | Identifier or catalogue number |
|---|---|---|
| clusterProfiler version 1.0 | Wu et al, 2021 | |
| Scanpy version 1.11.1 | Wolf et al, 2018 | |
| Seurat version 4.3.1 | Hao et al, 2021 | |
| MulEA version 1.1.1 | Turek et al, 2024 | |

## Source of SNP data

Existing Immunochip data from the Leuven IBD unit were used for the current analysis ($n = 1695$ Crohn's disease, $n = 941$ ulcerative colitis). All patients included in the analysis provided written informed consent to participate in the Leuven IBD Biobank, which was approved by the Institutional Review Board (B322201213950/S53684).

Demographics data of the patients is available in Appendix Table S1.

## SNP annotation

From the patient-specific genotype data, we identified non-coding SNPs associated with IBD and modelled the impact of these SNPs on regulatory regions of the genome, as we described earlier in Brooks-Warburton et al (Brooks-Warburton et al, 2022). Briefly, IBD-associated SNPs were first extracted from two landmark studies: Jostins et al. (Jostins et al, 2012) and Farh et al (Farh et al, 2015). Jostins et al. performed a meta-analysis of 15 IBD GWAS, followed by validation using Immunochip data from an independent cohort, identifying 163 distinct genome-wide significant genomic loci. To focus on the most likely causal variants, we selected those GWAS hits that overlapped with fine-mapped SNPs from Farh et al, who applied statistical fine-mapping based on linkage disequilibrium (LD) and functional annotations to identify candidate causal variants within each locus. Fine-mapped IBD-associated non-coding SNPs uniquely identified by Farh et al. were also selected to broaden the inclusion of high-confidence candidate regulatory variants.

From these SNP loci, we identified those that overlapped with known regulatory regions of the genome, i.e. promoters, enhancers, and miRNA-target sites. Promoters were defined as sites that are 5 kilobases (KB) upstream of the transcription start site (TSS) and downstream until the end of the first exon of the particular gene, according to the UCSC genome browser (Nassar et al, 2023). Enhancer regions were defined based on the HEDD database (downloaded on 13/08/2023) (Wang et al, 2018), which integrates experimental evidence from ENCODE (ENCODE Project Consortium et al, 2020), FANTOM5 (Abugessaisa et al, 2021), and the Epigenomics Roadmap (Roadmap Epigenomics Consortium et al, 2015). miRNA-target sites were identified using the miRanda tool (version 1.9) (Hsu et al, 2006). Note, heterozygous and homozygous genotypes were treated equivalently, as the iSNP pipeline is designed to capture whether a regulatory element can be perturbed at all rather than to model allele dosage.

The potential impact of SNPs located in enhancer or promoter regions on TF binding was assessed by: (i) collating known transcription factor binding sites (TFBSs) from the JASPAR database (date of download: 10/10/2023) (Castro-Mondragon et al, 2022) the Regulatory Sequence Analysis Tools (version 2018.8.1) (Medina-Rivera et al, 2015) and Find Individual Motif Occurrences tool (version 5.1.1) (Grant et al, 2011) to identify the likely TFs binding at genomic loci containing non-coding SNPs. TFBSs predicted to be perturbed by a regulatory SNP, either due to a gain or loss of binding site, were retained for downstream analysis.

SNP selection was further refined and contextualised based on epigenetic regulation by retaining only those overlapping with Chromatin Immunoprecipitation Sequencing (ChIP-seq) peaks located within tissues of interest. Three sources of ChIP-seq data were utilised: (1) tissue-specific histone methylation and acetylation ChIP-Seq data obtained from the ENCODE database (https://www.encodeproject.org/) (Qu and Fang, 2013), using pre-processed pseudoreplicated peak files (hg19 version) (see Dataset EV5 for details on tissues and files used); (2) H3K27ac ChIP-Seq profiles from intestinal epithelial samples of suspected IBD patients, retrieved from the Gene Expression Omnibus (https://www.ncbi.nlm.nih.gov/geo/) under accession number GSE51425 (Mokry et al, 2014); and (3) Cis-Regulatory Modules data for Homo sapiens (hg19 version) downloaded from the ReMap database (https://remap2022.univ-amu.fr/) (Hammal et al, 2022). This ensured that only SNPs located within experimentally validated, epigenetically active regulatory regions in intestinal tissues were retained.

In this way, a gene (and its protein product) was defined as "SNP-affected" when a fine-mapped IBD-associated non-coding SNP was predicted to perturb (i.e. create or abolish) a TFBS within an epigenetically active promoter or enhancer region, or a miRNA-target site.

## Network propagation algorithm with random seed control

SNP-affected proteins identified for each patient were selected as seeds to construct patient-specific SNP-propagated cellular signalling networks using a directed HotNet2 heat-propagation algorithm (Leiserson et al, 2015) from the implementation of Huang et al. (Huang et al, 2018), over a prior-knowledge network of human signalling interactions from OmniPath. In brief, the algorithm creates a kernel from the degree-weighted adjacency matrix with an alpha retain parameter, which describes how much heat is retained at each node (Eq. (1)).

$$K = (1 - \alpha)I \cdot (I - \alpha A_w)^{-1} \qquad (1)$$

Where K is the resulted kernel matrix, $\alpha$ is the retain parameter, I is the identity matrix and $A_w$ is the weighted adjacency matrix, where each edge is weighted according to the reciprocate of the out-degree of the source node.

$\alpha$ retain parameter was calculated using the following equation:

$$\alpha = m * \log 10 edgenum + b \qquad (2)$$

Here edgenum is the number of edges in the graph $m$ and $b$ are parameters, which were used as proposed by Huang et al ($m = -0.02935302$ and $b = 0.74842057$ (Huang et al, 2018).

A unit heat was propagated through the K kernel matrix based on Eq. (3).

$$H_{out} = HiK \qquad (3)$$

Here, the $H_i$ is a diagonal matrix with unit (1) value for heat at each SNP-affected protein, and K is the kernel. Note, if a patient has multiple SNPs, then the network will have a higher amount of heat to distribute. If the heat reached a TF, then it was propagated one step forward using the TF–TG network.

$$H_{real} = \frac{\sum_{TF} \frac{H_{TF}}{\#TG_{TF}}}{\#TG_{TF}} \qquad (4)$$

Here, $H_{TF}$ is the heat which reaches a transcription factor "TF" according to Eq. (3). $\#TG_{TF}$ is the number of targets of a transcription factor "TF". We sum the heat which proportionally affects each target gene "TG" from each "TF". $\#TF_{TG}$ is the number of transcription factors which affect target gene "TG".

To minimise citation bias of the network resources and avoid the overrepresentation of hubs, random seed controls were used. Here, for each patient, exactly the same number of seed genes were used to construct random SNP-propagated signalling and gene regulatory networks. After 10,000 repetitions from these random seeds, a distribution of heat was produced for each protein in the network for each patient. A Z-score was calculated from each distribution Eq. (5).

$$Z = \frac{H_{real} - \overline{H_{random}}}{SDH_{random}} \qquad (5)$$

Where $H_{real}$ is the heat of each node. $H_{random}$ is the heat vector distribution using the random control. SD is the standard deviation and $\overline{H_{random}}$ is the average of the random measures. If Z was >2 then the protein was significantly affected. The Z-score was similarly calculated in the case of TF-TG analysis. The used Z-score is a conservative estimation of affected nodes.

Thus, nodes that remained in the SNP-propagated signalling and gene regulatory networks had a 95% chance of representing non-random, SNP-driven perturbations, rather than arising from background network topology or citation bias.

The heat propagation by the kernel is analogous to a random walk with a restart method and represents a steady state (Cowen et al, 2017; Picart-Armada et al, 2019). Meanwhile, the regulatory one-step heat propagation represents one step of transcriptional response.

The analytical code is accessible in the GitHub repository of the Korcsmaros lab at https://github.com/korcsmarosgroup/iSNP.

## Network resources

Signalling interaction data were obtained from the OmniPath database (downloaded on 27/10/2021) (Türei et al, 2021, 2016). The directed network giant component was used, which consisted of 6,954 protein nodes and 51,792 edges. For TF-TG interactions, we used the DoRothEA database (downloaded on 27/10/2021) with confidence scores A, B, and C (Garcia-Alonso et al, 2019). This network contained 15,760 nodes and 90,145 edges.

To ensure that SNP effects were propagated to the most reliable TF–TG interactions, we restricted our analysis to regulons classified with A, B, or C confidence levels in DoRoThEA. This yielded 676 distinct human TFs or TF complexes. Of these, the vast majority (i.e., 517) were also represented in the OmniPath database, which we used to construct our signalling network. This overlap ensured that there was minimal potential for losing high-confidence TF–TG interactions during the two-step network propagation analysis.

Sensitivity analysis using other major signalling and gene regulatory network resources was performed. These included Reactome (downloaded on 23/03/2023) (Milacic et al, 2024), STRING (downloaded on 28/08/2023) (Szklarczyk et al, 2023), and CollecTRI (downloaded on 23/05/2025) (Müller-Dott et al, 2023). The STRING database was filtered for only experimental and database evidence scores and translated to UNIPROT IDs using the AnnotationDBI package (version 1.60.0) (Hervé Pagès, 2019). In addition, SNP-propagated networks were constructed using degree-matched random seeds to evaluate the robustness of the results to network topology.

## Overrepresentation analysis

For Reactome pathway overrepresentation analysis, we used the MulEA (version 1.1.1) (Turek et al, 2024) package in the R environment (version 4.2.2) (R Core Team, 2015). The background used was OmniPath in the case of protein–protein interaction network analysis and DoRothEA in the case of TF-TG analysis. The enrichment plots were visualised using the enrichplot package (version 1.18.3) (Yu, 2025). Identifier mapping was done using the org.Hs.eg.db (version 3.16.0) (Carlson, 2017) through AnnotationDBI (version 1.60.0) (Hervé Pagès, 2019).

## Visualisation and network analysis

Cytoscape (version 3.10.1) was used for visualisation (Shannon et al, 2003). In all networks, the network modules were created using the clusterMaker application (version 2.3.4) (Morris et al, 2011) GLay clustering, which is an implementation of the Grivan–Newman algorithm (Girvan and Newman, 2002). In short, this algorithm removes the highest betweenness edges from the network until the network falls into multiple components. These components are considered the new network modules.

This resulted in a network which better represented the patient-specific clusters. The networks are uploaded to the NDEX network resource (https://www.ndexbio.org/#/networkset/5618007c-5257-11ee-aa50-005056ae23aa?accesskey=3de7d45d693d83e2fd96029b0d856f27aaa6e82337193346bbfc4621a59cbd60).

## Patient-specific clustering

The patients were clustered using their SNP-propagated gene regulatory networks as a binary matrix as an input. First, a Jaccard distance was calculated (Jaccard, 1912) using SciPy pdist function (version 1.14.1) (Virtanen et al, 2020) in Python (version 3.11.5), then principal components were calculated using scikit-learn (version 1.5.1) pca function (Pedregosa et al, 2011). K-means clustering was used on this latent space, resulting in similarly sized clusters on the first three principal components, containing 95% of

variance, using scikit-learn's implementation. Hierarchical clustering only produced small clusters due to the presence of outlier patients in both diseases (Appendix Fig. S7). The optimal number of clusters was determined using the silhouette score (Rousseeuw, 1987) and the elbow method (Thorndike, 1953) implemented in the scikit-learn package (version 1.5.1) (Appendix Fig. S3) (Pedregosa et al, 2011).

## Single-cell RNA-seq comparison

For single-cell RNA-seq validation, the Kong et al dataset was used for CD (Kong et al, 2023). The dataset was downloaded from the Broad Institute Single Cell portal (Tarhan et al, 2023). The downloaded data was analysed by Seurat (version 4.3.1) (Hao et al, 2021) in an R environment. We considered only cells which were expressing at least 200 genes and had a mitochondrial gene count below 5% and genes which were present in at least three cells. The data was then log-normalised using Seurat's NormalizeData function using standard parameters and scaled using Seurat's ScaleData function using standard parameters. From the resultant Seurat object, using the FindMarkers function comparing cells originating from healthy and non-inflamed CD patients, DEGs were identified (Benjamini–Hochberg adjusted $P$ value < 0.1, | log2FC| > 0.5). These genes were used as the input for over-representation analysis comparing the cluster-representative genes in CD patients.

We also ran a similar analysis using another recently published CD single-cell RNA-seq dataset (Krzak et al, 2023). We downloaded the data from Zenodo (https://zenodo.org/records/105528153l). We used the standard Scanpy (version 1.11.1) (Wolf et al, 2018) pipeline in Python (version 3.11.5). Filtering was performed as per Krzak et al., with the following excluded: cells with fewer than 100 genes expressed at ≥1 or mitochondrial count >50, and genes expressed (>=1 count) in 5 or fewer cells. The counts were then normalised to $1 \times 10^5$, scaled and log transformed. We used the discovery cohort and the more granular cell type classification. We used Python DEseq2 (version 0.4.10) (Muzellec et al, 2023) pseudobulk analysis comparing CD patients with healthy controls. Only cell types with a minimum of 10 cells in each category and a minimum of 15 cells in total were used, and genes were filtered to have a minimum of 1000 counts. We defined DEGs from the pseudobulk results as Benjamini–Hochberg adjusted $P$ value < 0.05 and |log2FC| > 0.5. Note that the analysis of the two datasets uses slightly different pipelines to test the robustness of the outcome. We were unable to implement the same pipelines across the scRNAseq datasets as they were available in different formats, i.e., the Kong et al dataset was available as a Seurat object, whilst the Krzak et al dataset was available as a scanpy h5at file. Nevertheless, to maximise comparability, we analysed both datasets using DESeq2 in R and Python, respectively, and applied harmonised thresholds for differential expression i.e. |log2FC| of >= 0.5 and adjusted $P$ value (Benjamini–Hochberg) of <0.05.

We performed a similar analysis on the Smillie et al dataset, which was at the time, the most comprehensive single-cell RNA-seq dataset in UC (Smillie et al, 2019). However, we used slightly different filtering parameters due to the different quality of the dataset. Samples with less than 15% of mitochondrial transcript count and more than 200 genes, but less than 5000 expressed genes

were kept. Following this, we conducted the analysis with the exact same parameters as those applied to the Kong et al dataset.

## Pseudobulk analysis for transcription factor activity

Both the Kong et al and the Krzak et al datasets were used to validate the activity of TFs predicted to be perturbed in the iSNP pipeline. For the Kong et al analysis, we used a Seurat (version 4.3.1) pseudobulk pipeline comparing healthy controls with non-inflamed CD patients. We used the R DESeq2 (version 1.42.1) (Love et al, 2014) for differentially expressed gene analysis and kept all fold changes. After identifying DEGs, we mapped the HGNC gene names to UniProt Swiss-Prot IDs using the ENSEMBL Biomart package (version 2.58.2) (Smedley et al, 2009). We then ran a cell type-specific univariate linear model using the decoupleR package (version 2.9.7) (Badia-I-Mompel et al, 2022) to determine the activity of TFs from the pseudobulk differential expression analysis. DecoupleR infers TF activity by fitting a univariate linear model that evaluates whether the observed expression changes in the target genes of a particular TF are significantly associated with the known TF-TG interaction weights defined in DoRoThEA. This produces an activity score for each TF, which reflects the likelihood of functional perturbation.

In the case of Krzak et al, we ran the analysis in Python because the data was available in scanpy h5at format. The gene names were mapped to UniProt Swiss-Prot IDs through the pybiomart package (version 0.2.0) (https://jrderuiter.github.io/pybiomart/authors.html). Here, we ran the decoupler-py (version 1.7.0) (Badia-I-Mompel et al, 2022) tool from the pseudobulk outcome of the pydesq2 using the same DoRoThEA network as the input network as in the network propagation analysis and fitted a cell type-specific univariate linear model as described above.

## Enrichment analysis in the UNIFI cohort

The UNIFI induction trial was a randomised placebo-controlled study investigating the efficacy and safety of ustekinumab for the treatment of moderate-to-severe UC (Sands et al, 2019). Mucosal biopsies were collected at baseline from 364 patients who received ustekinumab (at 6 mg/kg or 130 mg intravenously) and 186 patients who received a placebo. Probe sets associated with multiple ENTREZ gene identifiers were discarded. Intensity data for probe sets mapped to the same ENTREZ gene identifier were summarised through their geometric mean. Differential gene expression analysis between responders ($n = 56$) versus non-responders ($n = 302$) was performed in R (R Core Team, 2015) using the limma package (version 3.46.0) (Ritchie et al, 2015). Response to therapy was assessed 8 weeks after induction. For this study, it was defined as mucosal healing, requiring endoscopic improvement (i.e., Mayo score ≤1) and histologic improvement (i.e., neutrophil infiltration in <5% of crypts, no crypt destruction, and no erosions, ulcerations, or granulation tissue).

## Transcriptomics validation

As part of the validation analysis, DEGs were obtained from analysing transcriptomics data from IBD and healthy controls (Dataset EV6). Using GEO2R (Barrett et al, 2013), the samples were normalised using limma (version 3.46.0) (Ritchie et al, 2015). DEGs

are defined by the following parameters: |log2FC| > 0.5 and adjusted $P$ value < 0.05 by Benjamini–Hochberg method. The following studies were used for comparison in CD: Vancamelbeke et al, 2017 (Vancamelbeke et al, 2017); Verstockt et al, 2019 (Verstockt et al, 2019); Pavlidis et al, 2021 (Pavlidis et al, 2021); and the following studies in UC: Arijs et al, 2009 (Arijs et al, 2009); Vancamelbeke et al, 2017 (Vancamelbeke et al, 2017); Van der Goten et al, 2014 (Van der Goten et al, 2014). The complete list of DEGs can be found on our GitHub page: https://github.com/korcsmarosgroup/iSNP. An overrepresentation test was conducted to see whether the SNP-affected genes, the genes in the SNP-propagated signalling network or SNP-propagated regulatory network were enriched in the DEGs using the clusterProfiler package (version 4.6.0) (Wu et al, 2021).

## Data availability

The computer code produced in this study are available in the following database: https://github.com/korcsmarosgroup/iSNP.

The source data of this paper are collected in the following database record: biostudies:S-SCDT-10_1038-S44320-025-00169-3.

## Peer review information

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

## Acknowledgements

DM and AB were funded by a European Research Council Starting Grant (336159). JPT is supported by the Chain Florey Clinical PhD Fellowship jointly funded by the National Institute for Health Research (NIHR) Imperial Biomedical Research Centre (BRC) and the UKRI Medical Research Council (MRC) Laboratory of Medical Sciences (LMS). DM acknowledges financial support from Imperial College London through an Imperial College Research Fellowship grant award. The work of DM, PS, SRC and TK was supported by the BBSRC Gut Microbes and Health Institute Strategic Programme BB/R012490/1 and its constituent project BBS/E/F/000PR10353. MP was supported by the UKRI Biotechnological and Biological Sciences Research Council (BBSRC) funded by Norwich Research Park Biosciences Doctoral Training Partnership (grant numbers BB/M011216/1 and BB/S50743X/1). DM, PS, TK, and SRC were also supported by a BBSRC Core Strategic Programme Grant for Genomes to Food Security (BB/CSP1720/1) and its constituent work packages, BBS/E/T/000PR9819 and BBS/E/T/000PR9817. SRC, TK and DM were supported in part by the UK Biotechnology and Biological Sciences Research Council (BBSRC) under grant numbers BB/J004529/1 and its constituent project BBS/E/F/000PR10355. BV is funded by the Clinical Research Fund (KOOR), University Hospitals, Leuven, Belgium and the Research Council KU Leuven. MM was founded through the UKRI BBSRC NRP DTP as a National Productivity Investment Fund CASE Award in collaboration with BenevolentAI (BB/S50743X/1). TK and SRC were also supported by the UKRI BBSRC Institute Strategic Programme Food Microbiome and Health BB/X011054/1 and its constituent project BBS/E/F/000PR13631. NP is supported by the Wellcome Trust (WT101159), Crohn's and Colitis UK. TK and NP were supported by the NIHR Imperial Biomedical Research Centre (BRC). The views expressed are those of the authors and not necessarily those of the NIHR or the UK Department of Health and Social Care.

## Author contributions

**Dezső Módos**: Conceptualisation; Resources; Data curation; Software; Formal analysis; Supervision; Investigation; Visualisation; Methodology; Writing—original draft; Project administration; Writing—review and editing. **John P Thomas**: Conceptualisation; Formal analysis; Investigation; Methodology; Writing—original draft; Writing—review and editing. **Johanne Brooks-Warburton**: Conceptualisation; Data curation; Investigation; Methodology; Writing—original draft; Writing—review and editing. **Martina Poletti**: Software; Formal analysis; Investigation; Visualisation; Methodology; Writing—review and editing. **Balazs Bohar**: Software; Formal analysis; Methodology; Writing—review and editing. **Yufan Liu**: Software; Formal analysis; Visualisation; Writing—review and editing. **Matthew Madgwick**: Software; Formal analysis; Methodology. **Luca Csabai**: Software; Formal analysis; Methodology. **Wen-Xin Kang**: Validation; Visualisation. **Benjamin Alexander-Dann**: Formal analysis; Investigation; Methodology. **Azedine Zoufir**: Formal analysis; Investigation; Methodology. **Padhmanand Sudhakar**: Software; Formal analysis; Investigation; Methodology. **Domenico Cozzetto**: Software; Formal analysis; Validation. **David Fazekas**: Software; Formal analysis; Investigation; Methodology. **Shamith Samarajiwa**: Data curation; Formal analysis. **Simon R Carding**: Resources; Supervision; Investigation. **Nicholas Powell**: Resources; Supervision; Funding acquisition; Investigation. **Bram Verstockt**: Resources; Data curation; Funding acquisition; Investigation; Methodology. **Andreas Bender**: Resources; Formal analysis; Supervision; Funding acquisition. **Tamas Korcsmaros**: Conceptualisation; Supervision; Funding acquisition; Investigation; Methodology; Writing—original draft; Project administration; Writing—review and editing.

Source data underlying figure panels in this paper may have individual authorship assigned. Where available, figure panel/source data authorship is listed in the following database record: biostudies:S-SCDT-10_1038-S44320-025-00169-3.

## Disclosure and competing interests statement

JBW, TK and SRC are named inventors on a patent application on the iSNP workflow to create disease-specific networks from SNP data. JPT has received research support from AstraZeneca. JBW has received lecture fees from Falk Pharma and financial support for research from AbbVie. DM got consultancy fees from HEALX and IOTA Pharmaceuticals. NP spoke for Allergan, Bristol Myers Squibb, Falk, Ferring, Janssen, Pfizer, Tillotts, and Takeda, and as a consultant and/or advisory board member for AbbVie, Allergan, Celgene, Bristol Myers Squibb, Ferring, and Vifor Pharma. BV has received research support from AbbVie, Biora Therapeutics, Landos, Pfizer, Sossei Heptares and Takeda; Speaker's fees from Abbvie, Biogen, Bristol Myers Squibb, Celltrion, Chiesi, Eli Lily, Falk, Ferring, Galapagos, Janssen, MSD, Pfizer, R-Biopharm, Takeda, Truvion and Viatris; Consultancy fees from Abbvie, Alimentiv, Applied Strategic, Atheneum, Biora Therapeutics, Bristol Myers Squibb, Eli Lily, Galapagos, Guidepoint, Landos, Mylan, Inotrem, Ipsos, Janssen, Progenity, Sandoz, Sosei Heptares, Takeda, Tillots Pharma and Viatris. The remaining authors declare no competing interests.

