## [Peer Review File · Molecular Systems Biology]

Decoding Non-coding SNPs: Systems Genomics Modelling Dissects the Heterogeneity of IBD

Dezso Modos, John Thomas, Johanne Brooks-Warburton, Martina Poletti, Balazs Bohar, Yufan Liu, Matthew Madgwick, Luca Csabai, Wen-Xin Kang, Benjamin Alexander-Dann, Azedine Zoufir, Padhmanand Sudhakar, Domenico Cozzetto, Dávid Fazekas, Shamith Samarajiwa, Simon Carding, Nick Powell, Bram Verstockt, Andreas Bender, and Tamas Korcsmaros

Corresponding author(s): Tamas Korcsmaros (t.korcsmaros@imperial.ac.uk)

Review Timeline:

Submission Date:	23rd Oct 24
Editorial Decision:	17th Dec 24
Appeal Received:	18th Sep 25
Editorial Decision:	11th Oct 25
Revision Received:	23rd Oct 25
Accepted:	27th Oct 25

Editor: Jingyi Hou

Transaction Report:

17th Dec 2024

RE: Manuscript MSB-2024-12721, "Decoding Non-coding SNPs: Systems Genomics Modelling Dissects the Heterogeneity of IBD"

Dear Dr Korcsmaros,

Thank you for submitting your work to Molecular Systems Biology. I would like to apologize for the somewhat slow process, which was due to the late arrival of reviewers' reports. We have now heard back from two of the three reviewers who agreed to evaluate your manuscript. Unfortunately, after a series of reminders, we did not manage to obtain a report from Reviewer #3. In the interest of time, I prefer to make a decision now rather than further delaying the process. As you will see below, the reviewers raise substantial concerns on your work, which unfortunately preclude its publication in Molecular Systems Biology.

While Reviewer #1 is relatively more positive, Reviewer #2 raised significant concerns about the general assumptions and approach of the study, pointing out several methodological and interpretational shortcomings that affect the study's validity and biological relevance. Specifically, Reviewer #2 rated the conceptual novelty, technical quality, validity of the conclusions, and suitability for publication as "low."

Given the substantial nature of the concerns raised, and in light of the relatively limited enthusiasm from the reviewers, we feel we have no option but to return the manuscript with the decision that we cannot offer publication at this time.

However, we would be open to considering a substantially revised and extended version of the manuscript, provided that the issues raised by the reviewers can be convincingly addressed. In particular, the points 2-6 raised by Reviewer #2 will need to be carefully addressed. I understand if, given the significant revisions required, you may prefer to submit your study elsewhere.

A resubmitted work would have a new number and receipt date. It will be editorially evaluated afresh, and its novelty will be re-assessed at the time of submission. As you probably understand, we can give no guarantee about its eventual acceptability. If you do decide to follow this course, then we would ask you to enclose with your re-submission a point-by-point response to the points raised in the present review.

I am sorry that the review of your work did not result in a more favorable outcome on this occasion, but I hope that you will not be discouraged from sending your work to Molecular Systems Biology in the future. In any case, thank you for the opportunity to examine this work.

Kind regards,
Jingyi

Jingyi Hou, PhD
Scientific Editor
Molecular Systems Biology

Reviewer #1:

Summary & General Remarks

The paper entitled "Decoding Non-Coding SNPs: Systems Genomics Modelling Dissects the Heterogeneity of IBD" presents a logical analysis of the effects of non-coding variants on downstream signalling and regulatory networks for two forms of IBD: Crohn's Disease (CD) and Ulcerative Colitis (UC). For each indication, disease-associated non-coding SNPs were identified from independent GWAS and linked to genes using the author's previously published iSNP procedure. These genes were then integrated with molecular networks, and the technique of network propagation used to identify neighbouring SNP-affected transcription factors and gene regulatory networks. These network "modules" were annotated based on functional enrichment, after which the gene regulatory networks were used to create patient clusters driven by specific TFs. It was further shown that patient clusters correspond to different cell type markers and TF activities in IBD-relevant tissue types. Finally, the authors performed a brief analysis to compare and contrast the identified signalling and regulatory networks across CD and UC,

highlighting shared pathways and distinct TFs.

The use of network propagation with individual patient data provides an interesting perspective on both global patterns and underlying heterogeneity of IBD driven by non-coding variation. As outlined in the introduction, this strategy addresses a key challenge for interpreting genetic associations for complex diseases such as IBD. The insights generated will be of interest to the IBD community, especially if additional details are provided for specific SNPs and mechanisms. The workflow presented could also be widely applied to other disease indications, though additional benchmarking could be useful to fully demonstrate methodological improvements. The results presented generally support the stated conclusions, though more details are needed in some results and methods sections to fully describe the results, analysis, statistical tests, and/or underlying data. Some conclusions rely heavily on functional enrichments that involve a small portion of network/module genes and may require further investigation. The manuscript is clearly written, and the figures are generally well-designed to convey the key messages. However, the use of black backgrounds could be reduced for easier viewing.

Specific comments are organized under numbered general headings, as follows:

1. Input Data & Definition of SNP-affected genes.

More details are needed to specify the OmniPath and Dorothea networks utilised. Downloading and filtering the Dorothea network via the R package (<https://saezlab.github.io/DoRothEA/>) gives a network of differing size, casting doubt on the results.

The ancestral makeup of the patient cohorts and input GWAS should be discussed. For SNP-affected genes, were heterozygous and homozygous genotypes considered equivalent?

A brief overview of the iSNP procedure should be provided in the methods. Does iSNP address non-additive effects? The network pipeline presented captures additive effects (polygenicity) but not genetic epistasis.

2. Construction of SNP-affected signalling and gene regulatory networks.

The authors acknowledge the concern of citation bias in the network resources. However, their random seed controls may not adequately account for the overestimation of hub genes in network propagation. Indeed, the identification of several very highly-studied TFs (e.g. TP53, NFKB1, STAT3) may be driven by such biases, and less-studied but more specific TFs may be missed. Degree-matched random seeds could be considered.

Can the authors further explain the rationale for having a two-step network analysis? Is there evidence that this approach performs better than propagation with a combined signalling and regulatory network? For the TF-TG networks, a local neighbourhood approach was used, resulting in the identification of thousands of targets for some TFs. A network propagation approach that captures the global topology of gene regulation may better prioritise the targets of pleiotropic TFs.

Based on Figure S1, a majority of SNP-affected genes are not present in the OmniPath signalling network. Have the authors explored that this may reduce the sensitivity of the study and potentially introduce biases driven by which SNP-affected genes are present in the signalling network? It would be useful to explore additional network resources and show that similar sets of transcription factors are prioritised.

Can the authors clarify the construction of SNP-propagated networks and patient-specific SNP-propagated networks? If the 38 seed genes are propagated, how are the "patient-specific networks" defined? If propagation is patient-specific, how are the networks in the figures defined?

Figure 1B implies that no propagation occurs beyond transcriptions factors, do all TFs have out-degree=0 in the signalling network?

How was the set of all transcription factors defined? And what proportion are present in the OmniPath and Dorothea networks?

3. Analysis of SNP-propagated signalling networks

It would be interesting to see the results of hierarchical clustering for the SNP-propagated signalling networks in Figures 2A & 5A. The network modules defined are large, and the functional enrichments presented involve only a small fraction of the module genes.

In Figure 2A (Also Figures 3A, 5A, 6A, 7A), can the authors clarify how module names were determined from the significantly enriched Reactome pathways in Supplementary Table 3 and address the discrepancy in the number of modules between the results (5) and table cover sheet (11)? Further, the legend states "CD patients grouped by Glrvan-Newman modularization" - should this be genes grouped by GN modularization?

In Figures 2A (Also Figures 3A, 5A, 6A & 7A), please clarify the definition of node size as a 'percent' or 'number' and provide a

quantitative scale.

Figure S2 is difficult to interpret. Further clarity is needed in describing the interpretation of box size, blank boxes etc.

4. SNP-propagated gene regulatory networks

Several different methods and thresholds are utilized for performing differential gene expression. The authors note that this is to "test the robustness of the outcome," but using different pipelines on the same dataset would be a better test of robustness.

What is the TF activity univariate linear model fitting? Are the models cell-type specific for both datasets or only the Krzak dataset?

Given the TF-centric nature of the gene regulatory networks in Figure 3A (and 6A), could they be simply represented as similarity networks between the transcription factors?

In validation analyses focusing on cluster-representative TFs, why are TFs such as FTL1 and ZFP36 not included?

In Figure 6E, how is 'Ustekinumab responder +/-' defined? Is there an explanation for why these designations do not align with the significant enrichments for response-predicting genes? Could patients with perturbations to the response predicting genes, in fact, be more sensitive to therapy?

5. Patient-specific results and clusters

The optimization procedure for determining the number of patient clusters is presented, but further justification for the selection of k is warranted. One could argue that the elbow method would better align with the selection of k=4 (CD) and k=5 (UC). Based on the following results, it also appears that the use of fewer clusters may give clearer separation between patient groups.

Given the pattern in Figure 6B doesn't conform to the expectation of a long-tailed distribution for frequencies, further dissection of the FOXP3 target peak is warranted to ensure that it is not due to an artifact. Is this pattern driven by a single SNP or multiple SNPs? Could there be artifacts from SNP analysis such as ref/alt designation, imputation, ancestry differences, or position on the X chromosome? Relatedly, in Figure 7C, it is surprising to see no FOXP3 interactions in the main CD patient grouping despite FOXP3 and its targets appearing in the CD SNP-propagated network.

Please clarify the terminology and provide definitions of patient-specific results, which are referred to as 'genes present in a patient', 'genes perturbed in patients', or 'genes predicted to be perturbed in patients'. Could the importance of genes be assessed statistically via comparison to random genes or non-IBD controls rather than as a count of affected patients?

Cluster-specific genes should be clearly defined in the main text and figures. How much overlap is there between genes assigned to each cluster?

Figure 3F (& 6F) - what is meant by 'representative examples'? If these represent single patients, why not average across all patients in each cluster?

6. Combined analysis of CD and UC

How were the combined CD/UC networks constructed? Did the authors compare combining the previously constructed networks to combining the SNP-affected genes prior to network analysis? Currently, the benefit of the network approach for distinguishing between CD and UC is unclear. It is stated that "STAT3 and TP53 were identified as TFs specifically perturbed in CD, whilst CEBPB, FOXP3, and NFKB1 were TFs specifically perturbed in UC," whereas ETS2 target genes were perturbed in both. However, from the iSNP results in Supplemental Table S2, these distinctions can already be made.

Could the "different disease-specific signaling interactions" and "distinct gene regulatory mechanisms" be the result of incomplete power in the UC and CD cohorts rather than true differences between the diseases? Support for similarities and differences could be developed by analysis of independent UC and CD GWAS cohorts.

In Figure S7, how is "disease-specific" defined for modules? Can this be quantified?

7. Discussion

The Supplementary Note contains many interesting details that could be included in the discussion. Currently, the main text contains few mechanistic interpretations of how non-coding SNPs influence IBD phenotypes, but these are critical for understanding the advances made in the study.

Reviewer #2:

This study aimed to develop a statistical framework for identifying pathways or functional modules associated with Crohn's Disease (CD) and Ulcerative Colitis (UC) through "SNP propagation" analysis. However, several significant methodological and interpretational flaws impact the study's validity and biological relevance.

Major Concerns:

1. The authors initiated their analysis using a set of "known IBD SNPs" identified in previous genome-wide association studies (GWAS). However, each GWAS-identified SNP is merely a proxy for an associated haplotype and does not necessarily have direct functional significance. Mapping individual SNPs onto transcription factor (TF) binding sites or microRNA (miRNA) target sites assumes biological roles that may not exist, potentially distorting the interpretation of GWAS findings.
2. The functional annotations for TF binding sites, promoters, enhancers, miRNA target sites and even the global network were ENTIRELY based on computational predictions. Numerous studies have demonstrated that these prediction-based genome annotations are highly prone to errors and lack reliability. Incorporating experimentally validated datasets, such as those from ENCODE and the Epigenome Roadmap, would enhance the study's robustness and interpretability. By relying solely on computational predictions, the authors have significantly weakened the biological credibility of their findings.
3. The authors claim to address cell-type specificity in their conclusions but fail to account for it meaningfully in their analyses. Predicted binding sites lack the resolution to capture tissue- or cell-specific epigenetic activity. For example, many SNPs may fall within binding sites that are inactive in the tissues relevant to CD and UC. Without specifying which binding sites are epigenetically active in the tissues of interest, the conclusions remain speculative. The authors should assess how many SNPs fall within functional sites that are confirmed as active in relevant tissues using epigenetic datasets.
4. The assumption that all SNPs located in functional TF or miRNA binding sites are biologically relevant is problematic. Most noncoding SNPs are likely functionally neutral. The study does not address whether these SNPs alter TF motifs, miRNA seed-complementary regions, or any other functional element in a meaningful way. Additionally, no experimental validation or in silico modeling of functional consequences was presented, leaving the claims unsubstantiated.
5. The study's GWAS dataset is notably underpowered compared to modern standards. The sample sizes for CD (1,695) and UC (941) are small relative to large-scale datasets such as those from the UK Biobank. As a result, the conclusions drawn from these data may lack statistical robustness. The study would benefit from replication using larger datasets to ensure that its findings are not spurious.
6. The downstream analyses are overly descriptive and lack statistical rigor, making it challenging to validate the conclusions. The text does not provide sufficient evidence or quantitative measures to support many of its claims. Without a more robust analytical framework, the study's conclusions remain speculative.

** As a service to authors, EMBO Press offers the possibility to directly transfer declined manuscripts to another EMBO Press title or to the open access journal Life Science Alliance launched in partnership between EMBO Press, Rockefeller University Press and Cold Spring Harbor Laboratory Press. The full manuscript and if applicable, reviewers' reports, are automatically sent to the receiving journal to allow for fast handling and a prompt decision on your manuscript. For more details of this service, and to transfer your manuscript please click on Link Not Available. **

Reviewer 1

1. More details are needed to specify the OmniPath and Dorothea networks utilised. Downloading and filtering the Dorothea network via the R package (<https://saezlab.github.io/DoRoThEA/>) gives a network of differing size, casting doubt on the results.

We thank the reviewer for kindly pointing this out. We have now added more details regarding the OmniPath and DoRoThEA networks we used including version number and date of download to ensure reproducibility of our findings.

2. For SNP-affected genes, were heterozygous and homozygous genotypes considered equivalent?

We thank the reviewer for this question. In our analysis using iSNP, heterozygous and homozygous genotypes were treated equivalently for SNP-affected genes. We have now stated this in the “Materials and Methods” section. This approach was chosen because iSNP is designed to capture whether a regulatory element can be perturbed at all, rather than to quantify allele dosage effects. While we acknowledge that homozygous variants could in theory have a stronger effect than heterozygous ones, the dosage effects of non-coding SNPs on TF binding has not yet been well characterised quantitatively and these may also vary across loci. We therefore considered the conservative approach of treating heterozygous and homozygous genotypes equivalently to be the most appropriate for this study. We have added this explanation to the “Discussion” section of the revised manuscript.

3. A brief overview of the iSNP procedure should be provided in the methods. Does iSNP address non-additive effects? The network pipeline presented captures additive effects (polygenicity) but not genetic epistasis.

Thank you for this suggestion - we have now provided an overview of the iSNP workflow in both the “Results” and “Materials and Methods” sections. We also appreciate the reviewer’s observation regarding additive versus non-additive effects. On reflection, we agree that the iSNP pipeline models polygenic contributions by integrating the impact of multiple fine-mapped non-coding SNPs on regulatory regions and downstream gene networks, rather than capturing non-additive interactions (genetic epistasis) between variants. We have therefore amended the Abstract, Introduction and Discussion sections accordingly.

4. The authors acknowledge the concern of citation bias in the network resources. However, their random seed controls may not adequately account for the overestimation of hub genes in network propagation. Indeed, the identification of several very highly-studied TFs (e.g. TP53, NFKB1, STAT3) may be driven by such biases, and less-studied but more specific TFs may be missed. Degree-matched random seeds could be considered.

Thank you for this insightful comment. We have now performed several analyses to address the possibility of citation bias in our results.

Firstly, we increased the number of random seed controls from 1,000 to 10,000 iterations to more accurately construct SNP-propagated networks relative to the size of the background network resources used. This allowed us to more accurately

remove noise from the SNP-propagated networks and better account for hub nodes, as evidenced by the lower number of SNP-affected proteins and TF-target genes that passed the threshold of statistical significance in both the SNP-propagated signalling and gene regulatory networks after 10,000 iterations compared to 1,000. This is now properly presented in the revised “Materials and Methods” section.

Secondly, we performed comprehensive comparative analyses using three major signalling networks i.e. OmniPath, Reactome, STRING, and two complementary regulatory network resources DoRoThEA, and Collectri as now stated in the “Materials and Methods” section. We observed differences in the number of perturbed proteins in the SNP-propagated signalling network and perturbed TF-target genes in the SNP-propagated gene regulatory network (GRN) based on the background networks used which have been described in the Discussion and depicted in Figure S8.

Thirdly, following the reviewer’s suggestion, we constructed networks using degree-matched random seeds. These analyses identified more perturbed proteins and TF–target genes than the non-degree-matched controls. However, confirming the original results, all cases (irrespective of the underlying network resources used; i.e. OmniPath, Reactome, STRING, DoRoThEA, and Collectri) included the core set of perturbed nodes identified in the non-degree matched random seed networks. This indicates that degree-matched random seeds propagate SNP signals more broadly. We therefore chose to report the non-degree-matched results in our main results, as they capture the consistently identified proteins and TF–target genes in a more stringent and reproducible manner. We have added the degree matched results to the revision (Fig S11) and we added this comparison to “Materials and Methods” and “Discussion” section of the revised manuscript.

5. Can the authors further explain the rationale for having a two-step network analysis? Is there evidence that this approach performs better than propagation with a combined signalling and regulatory network? For the TF-TG networks, a local neighbourhood approach was used, resulting in the identification of thousands of targets for some TFs. A network propagation approach that captures the global topology of gene regulation may better prioritise the targets of pleiotropic TFs.

We thank the reviewer for pointing out that this methodology step was not described fully in the original manuscript.

We adopted a two-step network propagation analysis to separately capture the molecular mechanisms likely to be impacted by SNPs at the signalling layer and at the direct downstream gene regulatory layer. This separation allows us to extract complementary and biologically meaningful insights into UC and CD pathogenesis that may be obscured in a single combined network, where information from distinct interaction types can be collapsed.

To our knowledge, direct benchmarking of two-step versus combined network propagation in complex diseases has not yet been reported. However, related work outside of this context supports the value of the two-step, multilayered approach. For example, TieDIE (Paull et al., 2013) applies separate diffusion processes from signalling inputs and regulatory outputs (such as transcription factors) and identifies convergence points, achieving higher precision in cancer datasets compared with

single diffusion on combined networks. Importantly, TieDIE demonstrated that separating diffusion reduces bias toward highly connected nodes, which in a combined network tend to dominate the propagation and obscure disease-specific mechanisms. Similarly, Valdeolivas et al. (2019) showed that random walks on multiplex/heterogeneous networks outperform those on monoplex or aggregated networks, as they preserve the complementary contributions of each interaction type without loss of information from collapsing layers. Together, these findings suggest that multilayer propagation better captures specific links between genetic alterations and transcriptional consequences, rather than disproportionately prioritising generic hubs. We have added these references and explanation to the “Discussion” section of the revised manuscript.

To construct the SNP-propagated gene regulatory network, we intentionally utilised a one-step (local) heat propagation as this better reflects the broader, more promiscuous interactions that are known to occur between TFs and their targets in a whole gene regulatory network - where the SNP effects would be diluted due to network distance. This decision was further validated by our analysis of the Kong et al single cell transcriptomics dataset, where we found that patient-specific clusters of SNP-propagated gene regulatory modules (based on one-step heat propagation) mapped to differential gene expression in a cell type-specific manner in CD. However, we acknowledge that global propagation methods, which leverage the full gene regulatory network topology, could in future provide a complementary method to prioritise targets of pleiotropic TFs. We have extended the “Discussion” section with this explanation.

6. Figure 1B implies that no propagation occurs beyond transcriptions factors, do all TFs have out-degree=0 in the signalling network?

Thank you for kindly pointing this out. TFs can interact with each other and other regulatory proteins, so they can have an out-degree of >0 . We have not excluded these interactions, but we appreciate that the original Figure 1B was misleading. So we have now included in the same panel (Figure 1C in the revised manuscript), non-TFs where the heat also converges for clarity and also TFs interacting with each other.

7. How was the set of all transcription factors defined? And what proportion are present in the OmniPath and Dorothea networks?

The set of all transcription factors (TFs) was defined from DoRoThEA, and we used only regulons classified with A, B, or C confidence levels. We added the details of this step to the “Materials and Methods” section of the revised manuscript - and we thank the reviewer for pointing out that this was missing. To ensure that SNP effects were propagated to the most reliable TF–TG interactions, we restricted our analysis to regulons classified with A, B, or C confidence levels in DoRoThEA. This yielded 676 distinct human TFs or TF complexes. Of these, the vast majority (i.e. 517) were also represented in the OmniPath database, which we used to construct our signalling network. This overlap ensured that there was minimal potential for losing high-confidence TF–TG interactions during the two-step network propagation analysis.

8. It would be interesting to see the results of hierarchical clustering for the SNP-propagated signalling networks in Figures 2A & 5A.

Thank you for this suggestion. We have now added these as Figure S7.

9. In Figure 2A (Also Figures 3A, 5A, 6A, 7A), can the authors clarify how module names were determined from the significantly enriched Reactome pathways in Supplementary Table 3 and address the discrepancy in the number of modules between the results (5) and table cover sheet (11)? Further, the legend states "CD patients grouped by Girvan-Newman modularization" - should this be genes grouped by GN modularization?

We thank the reviewer for these helpful comments. All significantly enriched Reactome pathways for the various networks are provided in Supplementary Tables 3. For the sake of visual clarity, we selected and displayed only the most IBD-relevant signalling and gene regulatory pathways in Figures 2A, 3A, 5A, 6A, and 7A. We have now clarified this in the relevant figure captions.

Regarding the discrepancy in the number of modules in the SNP-propagated signalling layer, we have now clarified in the Results section that we focus on the four largest modules in the text, whereas Table S3 includes all modules identified, including smaller ones.

Finally, we thank the reviewer for noting the inaccurate legend wording - we have corrected this to "genes grouped by Girvan–Newman modularization" rather than "CD patients".

10. Based on Figure S1, a majority of SNP-affected genes are not present in the OmniPath signalling network. Have the authors explored that this may reduce the sensitivity of the study and potentially introduce biases driven by which SNP-affected genes are present in the signalling network? It would be useful to explore additional network resources and show that similar sets of transcription factors are prioritised.

We thank the reviewer for this important comment. We have now conducted a broad benchmark analysis to justify and present the effect of various network resources on the analysis. We added this as Figure S1, S8 and S9. and presented it in the "Discussion" section of the revised manuscript. Out of the 95 possible SNP-affected proteins in UC, 51 were found in OmniPath. Out of the 47 SNP-affected proteins in CD, 30 were found in OmniPath. In both cases, OmniPath provided the largest coverage of SNP-affected proteins in comparison to STRING and Reactome, which is why we opted to use OmniPath. Our analyses, as described earlier in response to question 4, also demonstrate that the selection of network resources can influence propagation outcomes. We have now stated this in our "Discussion" and "Materials and Methods" sections.

11. In Figures 2A (Also Figures 3A, 5A, 6A & 7A), please clarify the definition of node size as a 'percent' or 'number' and provide a quantitative scale.

Thank you for this suggestion. We have now added a scale for node size proportional to percentage total patients in the figures of the revised manuscript.

12. Figure S2 is difficult to interpret. Further clarity is needed in describing the interpretation of box size, blank boxes etc.

Thank you, we agree that this supplementary figure was difficult to interpret and not informative, and have therefore now removed it from the manuscript.

13. Several different methods and thresholds are utilized for performing differential gene expression. The authors note that this is to "test the robustness of the outcome," but using different pipelines on the same dataset would be a better test of robustness.

We thank the reviewer for this suggestion. Unfortunately, we were unable to implement the same pipelines across the scRNAseq datasets as they were available in different formats i.e. the Kong et al dataset was available as a Seurat object, whilst the Krzak et al dataset was available as a scanpy h5at file. Nevertheless, to maximise comparability, we analysed both datasets using DESeq2 in R and Python respectively, and applied harmonised thresholds for differential expression i.e. $|\log_2FC|$ of ≥ 0.5 and adjusted p-value (Benjamani-Hochberg) of <0.05 . We have now revised the manuscript text to make this challenge clearer in the Methods section. For bulk microarray transcriptomics datasets we used the standard limma pipeline.

14. What is the TF activity univariate linear model fitting? Are the models cell-type specific for both datasets or only the Krzak dataset?

We thank the reviewer for these questions. We used decoupleR to provide TF activity scores from the cell type-specific differential expression analysis results. This TF activity is inferred by fitting a univariate linear model that evaluates whether the observed expression changes in the target genes of a particular TF are significantly associated with the known TF–target gene interaction weights defined in DoRoTheA. This produces an activity score for each TF, which reflects the likelihood of functional perturbation. As we performed this analysis using each of the two scRNAseq datasets, the models are cell type-specific for both datasets. We have now revised the text relating to this in the “Materials and Methods” to make these points clearer.

15. In validation analyses focusing on cluster-representative TFs, why are TFs such as FTL1 and ZFP36 not included?

We thank the reviewer for this helpful comment. In the validation analyses, we have now incorporated all transcription factors that were affected in the network and were significantly differentially regulating based on the single cell RNA-seq using the decoupleR univariate linear model.

16. In Figure 6E, how is 'Ustekinumab responder +/-' defined? Is there an explanation for why these designations do not align with the significant enrichments for response-predicting genes? Could patients with perturbations to the response predicting genes, in fact, be more sensitive to therapy?

Thank you for these questions. We hypothesise that if there is a significant overlap between DEGs that are associated with treatment response and those that are perturbed in the SNP-propagated gene regulatory network, then this may lower the chance of response to treatment. However, we agree that since iSNP does not model the directionality of these perturbations, there is also a possibility that these may also confer increased sensitivity to therapy. Therefore, these patient modules are likely to demonstrate differential responses to ustekinumab therapy, but directionality of these

differential responses are currently unknown. Hence, we have removed the “Potential to response” row in Figure 6E and kept only the statistics with regards to enrichment of DEGs comparing ustekinumab response vs non-response, and have also tempered the wording in our Results and Discussion sections accordingly.

17. The optimization procedure for determining the number of patient clusters is presented, but further justification for the selection of k is warranted. One could argue that the elbow method would better align with the selection of $k=4$ (CD) and $k=5$ (UC). Based on the following results, it also appears that the use of fewer clusters may give clearer separation between patient groups.

We selected the optimal number of clusters using both the silhouette score and the elbow method, as implemented in the scikit-learn package. Specifically, we chose the number of clusters that maximised the average silhouette score, while also considering the point at which the within-cluster sum of squares (WCSS) showed diminishing returns, as indicated by the elbow method. These complementary metrics guided cluster selection, but we agree with the reviewer that there is some inherent degree of subjectivity when defining biologically meaningful patient groups which cannot be fully captured by quantitative metrics alone. We added this point to the “Discussion” section of the revised manuscript.

18. Given the pattern in Figure 6B doesn't conform to the expectation of a long-tailed distribution for frequencies, further dissection of the FOXP3 target peak is warranted to ensure that it is not due to an artifact. Is this pattern driven by a single SNP or multiple SNPs? Could there be artifacts from SNP analysis such as ref/alt designation, imputation, ancestry differences, or position on the X chromosome? We thank the reviewer for this insightful comment. Figure 6B does not conform to an expected long tailed or scale free distribution, because the peaks reflect hub transcription factors which appear to regulate target genes that are affected in many patients in our cohort. FOXP3 showed the highest number of target genes ($n = 1905$) compared with other major TFs such as NF κ B1 ($n = 361$) and CEBPB ($n = 405$) - we have now annotated these peaks on Figure 6B. Importantly, these hub transcription factors are identified through the network propagation approach implemented in iSNP, which captures the cumulative effects of multiple non-coding SNPs converging on regulatory networks, rather than being driven by a single SNP or by direct SNP–TF effects.

19. Could the importance of genes (i.e. perturbed in patients) be assessed statistically via comparison to random genes or non-IBD controls rather than as a count of affected patients?

Following the reviewer suggestion, we generated a null distribution of 10,000 control networks generated from random seeds (see Materials and Methods) for the SNP-propagated networks, which allowed us to identify nodes and edges that passed a threshold of statistical significance compared to the null distribution. This is now presented in the “Materials and Methods” and “Discussion” sections of the revised manuscript.

20. Cluster-specific genes should be clearly defined in the main text and figures. How much overlap is there between genes assigned to each cluster?

Thank you for this comment. We realise that “cluster-representative genes” would be a more accurate term than “cluster-specific genes”, as these genes are not necessarily specific to a cluster. Instead, cluster-representative genes are those that

occur in more than 50% of patients in a particular cluster. We have now defined these clearly in the main text and figures.

21. Figure 3F (& 6F) - what is meant by 'representative examples'? If these represent single patients, why not average across all patients in each cluster?

Thank you for this great suggestion. We have now replaced figures 3F and 6F with average networks of all patients in each cluster.

22. Currently, the benefit of the network approach for distinguishing between CD and UC is unclear. Could the "different disease-specific signaling interactions" and "distinct gene regulatory mechanisms" be the result of incomplete power in the UC and CD cohorts rather than true differences between the diseases?

We thank the reviewer for pointing out that the power of our approach in comparing CD and UC was unclear. The SNP-propagated signalling and gene regulatory networks for UC and CD were derived from the non-coding SNPs present in each patient in our respective cohorts. Although only six non-coding SNPs overlapped between UC and CD patients - and every patient carried a unique SNP combination - our comparative analysis revealed that UC and CD SNPs converge on many of the same signalling pathways (albeit through different network routes) while largely perturbing distinct gene regulatory modules. These results highlight the ability of the systems genomics approach implemented in iSNP to detect the cumulative influence of multiple non-coding SNPs across both signalling and regulatory layers. While we agree that increasing the cohort size could further enhance power, the current cohort was sufficiently diverse to reveal shared signalling patterns alongside disease-specific regulatory mechanisms, supporting the robustness of our findings. We have now expanded on the value of this comparative analysis in the Discussion section of the revised manuscript.

23. Support for similarities and differences could be developed by analysis of independent UC and CD GWAS cohorts.

We agree with the reviewer that analysis of independent IBD GWAS cohorts could further support similarities and differences between CD and UC, which we have now highlighted in the "Discussion" section as a potential future work.

24. In Figure S7, how is "disease-specific" defined for modules? Can this be quantified?

On this supplementary figure (Figure S6A in the revised manuscript), a disease-specific module was defined if all the genes belonging to a certain module were affected only in UC or CD.

25. The insights generated will be of interest to the IBD community, especially if additional details are provided for specific SNPs and mechanisms.

We thank the reviewer for this statement and suggestion. In response, we have substantially expanded the “Discussion” section as recommended and incorporated specific mechanisms previously presented in the Supplementary Discussion into the main text.

26. However, the use of black backgrounds could be reduced for easier viewing.

We thank the reviewer for this suggestion. While we agree that black backgrounds may be less suitable for printed material, feedback from colleagues indicated that black backgrounds provide better contrast and visibility for viewing the networks with a high number of colour-coded information on digital screens.

We are grateful to the Reviewer for their detailed evaluation of our manuscript and for providing constructive suggestions to enhance our method, its presentation, and discussion. We trust that our revisions have now clarified and substantiated the points that were previously unclear or insufficiently supported.

Reviewer 2

1. The authors initiated their analysis using a set of "known IBD SNPs" identified in previous genome-wide association studies (GWAS). However, each GWAS-identified SNP is merely a proxy for an associated haplotype and does not necessarily have direct functional significance. Mapping individual SNPs onto transcription factor (TF) binding sites or microRNA (miRNA) target sites assumes biological roles that may not exist, potentially distorting the interpretation of GWAS findings.

We thank the reviewer for this important comment, and for pointing out that our method description was incomplete, which could lead to misinterpretation of our study. To address the comment, we did not rely solely on GWAS lead SNPs but instead focused on fine-mapped variants that are more likely to represent causal alleles. Specifically, we used IBD-associated SNPs from two landmark studies:

- Jostins et al. (2012): a meta-analysis of 15 GWAS, validated in an independent ImmunoChip cohort, identifying 163 genome-wide significant loci.
- Farh et al. (2015): statistical fine-mapping of GWAS loci incorporating LD structure and functional annotations to identify candidate causal variants.

From these datasets, we selected fine-mapped IBD-associated SNPs and further filtered for those overlapping with experimentally validated regulatory regions. This strategy allowed us to prioritise variants with the highest likelihood of regulatory function. We have now added these details to the Methods section.

We agree with the reviewer that some false positives or negatives may remain, as fine-mapping cannot definitively prove causality. However, resolving functional effects at each locus would require extensive experimental validation, which is beyond the scope of this study. Our goal here was to build a disease-relevant, high-confidence variant set to model potential downstream signalling and regulatory consequences, thereby providing a framework that can guide future functional genomics work and also aid precision medicine efforts. We have now clarified the limitation of our approach in the "Discussion" section of the revised manuscript.

2. The functional annotations for TF binding sites, promoters, enhancers, miRNA target sites and even the global network were ENTIRELY based on computational predictions. Numerous studies have demonstrated that these prediction-based genome annotations are highly prone to errors and lack reliability. Incorporating experimentally validated datasets, such as those from ENCODE and the Epigenome Roadmap, would enhance the study's robustness and interpretability. By relying solely on computational predictions, the authors have significantly weakened the biological credibility of their findings.

We thank the reviewer for this insightful comment. We would like to clarify that our pipeline does incorporate experimentally validated datasets, ensuring that our analyses are grounded in experimental evidence. As now stated in the "Materials and Methods" section of the revised manuscript, enhancer regions were defined based on data from multiple experimentally validated resources, including ENCODE, and the Epigenomics Roadmap.

To further contextualise our findings in IBD, encouraged by the reviewer's expert comment, we have now additionally integrated Chromatin Immunoprecipitation Sequencing (ChIP-Seq) data from intestinal tissues, drawing from the ENCODE database, the ReMap database, and a published dataset from IBD patients. These datasets which capture TF binding and histone modification patterns, allowed us to focus on non-coding SNPs located within experimentally supported regulatory genomic regions active in the primary tissue of interest. We have added these details to both the "Results" and "Materials and Methods" sections of the revised manuscript.

3. The authors claim to address cell-type specificity in their conclusions but fail to account for it meaningfully in their analyses. Predicted binding sites lack the resolution to capture tissue- or cell-specific epigenetic activity. For example, many SNPs may fall within binding sites that are inactive in the tissues relevant to CD and UC. Without specifying which binding sites are epigenetically active in the tissues of interest, the conclusions remain speculative. The authors should assess how many SNPs fall within functional sites that are confirmed as active in relevant tissues using epigenetic datasets.

We thank the reviewer for this important comment - we agree that the original analysis lacked support of tissue specificity in our analysis. As mentioned in the previous response, to address tissue- and disease-specific activity, we have now incorporated Chromatin Immunoprecipitation Sequencing (ChIP-Seq) data from intestinal tissues, drawing from the ENCODE and ReMap databases, as well as a published dataset from suspected IBD patients. This enabled us to retain only IBD-associated non-coding SNPs located within epigenetically accessible regulatory regions relevant to the primary tissue and disease context of interest.

4. The assumption that all SNPs located in functional TF or miRNA binding sites are biologically relevant is problematic. Most noncoding SNPs are likely functionally neutral. The study does not address whether these SNPs alter TF motifs, miRNA seed-complementary regions, or any other functional element in a meaningful way. Additionally, no experimental validation or *in silico* modeling of functional consequences was presented, leaving the claims unsubstantiated.

We thank the reviewer for this comment and agree that careful consideration of functional consequences is essential. We acknowledge the original manuscript did not make it clear that iSNP already incorporated a comprehensive *in silico* modelling framework using several complementary approaches. These were detailed in the first publication of the pipeline (Brooks-Warburton et al, Nature Communications, 2022), but we agree with the reviewer that more details in this work could clarify and improve the confidence in the current analysis. In particular we used two computational steps to specifically model the effect of non-coding SNPs:

- i) Transcription factor (TF) motifs: iSNP utilised FIMO and RSAT tools to scan genomic loci for TF binding motifs using position weight matrices (PWMs) derived from experimentally validated TF binding data.

- ii) miRNA target sites: iSNP used the miRanda tool to assess whether non-coding SNPs alter seed region complementarity and binding affinity at predicted miRNA target sites.

We have now expanded the “Methods” section to explicitly describe these steps, which were previously referenced but not detailed in the manuscript.

As an additional response to the reviewer’s comment, we have now validated the biological relevance of our pipeline by showing that SNP-propagated gene regulatory networks were enriched for differentially expressed genes between CD patients and non-IBD controls in a cell type–specific manner, using two independent scRNA-seq datasets. This confirms that the pipeline effectively captures the cumulative functional consequences of genetic variation on gene regulation and downstream expression. Complementing these results, all 8 of the major cluster-representative transcription factors identified through the iSNP pipeline were predicted to have perturbed activities in the Kong *et al* dataset (and 7 out of 8 in the Krzak et al dataset) using decoupleR.

In the “Discussion” section, we have also acknowledged that experimental validation of individual TF motif or miRNA seed binding changes due to non-coding SNPs was beyond the scope of this study. Rather, the aim of this work was to develop a novel systems genomics workflow that can be applied in complex diseases to identify candidate cellular signaling and gene regulatory mechanisms that could be influenced by modelling the downstream, cumulative effects of non-coding SNPs, thereby providing a resource to guide future experimental studies and precision medicine approaches. The complementary *in silico* modelling approaches we have implemented substantially enhance the credibility and interpretability of our findings in IBD.

5. The study's GWAS dataset is notably underpowered compared to modern standards. The sample sizes for CD (1,695) and UC (941) are small relative to large-scale datasets such as those from the UK Biobank. As a result, the conclusions drawn from these data may lack statistical robustness. The study would benefit from replication using larger datasets to ensure that its findings are not spurious.

Thank you for this comment highlighting a misunderstanding stemming from unclear method definition in the original manuscript. We would like to clarify that this study is not a GWAS: instead, we utilised high-confidence SNPs that had already been robustly identified in landmark IBD GWAS and fine-mapping studies (Jostins et al.; Farh et al.), to predict the downstream consequences of these SNPs on cellular signalling and gene regulatory networks.

The reviewer is fully correct that a cohort of 2,636 patients would not provide sufficient power to detect novel SNP associations in UC and CD patients using a conventional GWAS framework. Our approach builds on previous GWAS and extends our insights into the findings of those well-powered studies. We have now clarified this in the "Introduction" and "Discussion" sections of the revised manuscript.

6. The downstream analyses are overly descriptive and lack statistical rigor, making it challenging to validate the conclusions. The text does not provide sufficient evidence or quantitative measures to support many of its claims. Without a more robust analytical framework, the study's conclusions remain speculative.

We thank the reviewer for this comment. We would like to clarify that our downstream analyses are supported by a robust statistical framework, which we have now described more explicitly and extended with additional, new statistical analysis in the "Methods" and "Results" sections of the revised manuscript:

- I. Permutation testing - We have now performed permutation testing with random seeds 10,000 times (increased from 1,000 iterations in the original analysis) to generate control networks. This allowed us to more robustly identify perturbed SNP-propagated proteins and TF-target gene interactions that surpassed significance thresholds (z-score > 2 in at least one patient)
- II. Over-representation analysis - We performed over-representation analysis (ORA) on genes affected in more than 100 patients, applying a false discovery rate (FDR) threshold of < 0.1 to determine significant pathway enrichment.
- III. Patient-specific clustering - Jaccard and k-medoid based clustering of SNP-propagated regulatory networks was performed. The silhouette score and elbow method was utilised to determine the optimal number of clusters.
- IV. Single-cell RNA sequencing analysis - analysis of pseudobulked single-cell transcriptomics (scRNAseq) data was performed using the thresholds of $|\log_2FC|$ of ≥ 0.5 and adjusted p-value (Benjamani-Hochberg) of <0.05 for differential gene expression.
- V. Bulk transcriptomics analysis - was analysed using limma with the differential expression threshold set to $|\log_2FC|$ of ≥ 0.5 and adjusted p-value (Benjamani-Hochberg) of <0.05.

We are grateful to the Reviewer for their critical assessment of our work. The comments raised were highly relevant, highlighting areas where additional analysis, detail or clarification was needed. We believe that our revisions and responses have now addressed these concerns and provide sufficient support for our study.

Reviewer 3

1. The manuscript introduces a novel systems genomics workflow that integrates network propagation to elucidate the downstream effects of non-coding SNPs on signaling and gene regulatory pathways. This is methodologically robust and adds significant value to IBD research, presenting a strong systems genomics framework to understand non-coding SNPs in IBD, with novel insights into gene regulatory mechanisms and patient stratification.

We are pleased and grateful for the reviewer to point out the methodological robustness and significance of our work.

2. While computational predictions are compelling, experimental validation of the identified pathways and transcription factors is limited. For example, validating the role of STAT3 and FOXP3 in patient clusters through *in vitro* or *in vivo* studies would greatly strengthen the manuscript. Add examples of functional validation or at **least propose clear experimental follow-ups** for the most significant findings can improve the manuscript.

We thank the reviewer for this important comment and their understanding that experimental functional analyses would be outside the scope of the current manuscript. One way we have already validated the role of major transcription factors, such as STAT3, in our patient clusters was through transcription factor activity inference from single-cell transcriptomics data of Crohn's disease patients, as presented in the "Results" section. Several experimental follow-ups could further strengthen these findings encompassing multi-omics validation, *in vitro* functional perturbation, and *in vivo* testing, which we have now been added to the "Discussion" section of the revised manuscript.

3. While clusters were identified using SNP-propagated gene regulatory networks, their biological or clinical implications are not fully explored. For instance, why some transcription factors dominate specific clusters remains unclear. The authors might want to provide deeper biological interpretation or discuss the relevance of the dominant transcription factors in the identified clusters, or at least propose future studies in this direction.

We thank the reviewer for this thoughtful comment. In the iSNP pipeline, network propagation integrates and propagates the effects of multiple disease-associated non-coding SNPs, which converge onto transcription factors. As a result, transcription factors with larger regulons (such as FOXP3) are more likely to dominate individual patient clusters. Our computational analyses provided evidence for the functional relevance of these regulons and suggested their likely cell type-specific contexts. For instance, we found that only UC patient clusters containing gene regulatory modules regulated by NFKB1 were enriched for DEGs distinguishing ustekinumab responders from non-responders, indicating NFKB1-driven regulation may be a potential determinant of ustekinumab response. In addition, we identified ETS1 and ETS2 - the latter recently reported by Stankey et al, (Nature 2024) as a master regulator in CD - as the only transcription factors perturbed by non-coding SNPs in both UC and CD. We have added these clinical and biological insights to the "Discussion". Nonetheless, we agree that further biological interpretation is important, and deeper mechanistic insights could be obtained through the experimental strategies outlined in the previous response.

4. minor comments: Figures like 2 and 5 could be more readable by providing zoomed-in views for key modules or improve figure legends to clarify details such as node size and colors. This is a valid visualisation comment. However, now the revised network figures in the updated manuscript contain larger node names, as the size of the signalling networks have shrunk significantly after using a greater number of random seeds, resulting in a less noisy network. We added to the edge colours as main module specific edges.

We are grateful to the Reviewer for acknowledging the importance and robustness of our work, while constructively suggesting additional information to enhance its impact. By incorporating these details to better guide and support future studies, we trust that our revisions and responses now provide clearer and more valuable insights for both the Reviewer and future readers of this paper.

11th Oct 2025

Manuscript Number: MSB-2024-12721R-Q

Title: Decoding Non-coding SNPs: Systems Genomics Modelling Dissects the Heterogeneity of IBD

Author: Dezso Modos

John Thomas

Johanne Brooks-Warburton

Martina Poletti

Balazs Bohar

Yufan Liu

Matthew Madgwick

Luca Csabai

Wen-Xin Kang

Benjamin Alexander-Dann

Azedine Zoufir

Padhmanand Sudhakar

Domenico Cozzetto

Dávid Fazekas

Shamith Samarajiva

Simon Carding

Nick Powell

Bram Verstockt

Andreas Bender

Tamas Korcsmaros

Dear Tamas,

Thank you for submitting your revised manuscript. As the original Reviewers #1 and #2 were unavailable for this round, we have engaged a new Reviewer (Reviewer #4) to assess your responses to their previous comments. As you will see below, this reviewer is overall supportive. However, before we can accept this manuscript for publication, a few remaining issues raised still need to be addressed.

In addition, at a more editorial level, please do the following:

1. Reduce the keyword number to five.
2. Remove the "Authors' contribution" section from the manuscript file.
3. All figures should be removed from the main manuscript file. The figure legends need to be retained and placed in a separate section after the Reference list.
4. On the title page, please clearly denote the corresponding author(s) within the author list.
5. "Bibilography" should be renamed to "References".
6. Figure callouts: all callouts should be listed sequentially. Please add the missing callouts for Fig. 1A-E and 2C.
7. Please ensure the funding information in the manuscript matches the details provided in the submission system. The following grant is currently missing from the manuscript and must be added: UKRI | Biotechnology and Biological Sciences Research Council (BBSRC), grant number BBS/E/F/000PR10355.
8. For Tables S2-S4 and Tables S6-S8: Source file names, titles, legends and manuscript callouts all need to be updated to Dataset EV1-EV#, since they are rather large. The remaining Tables Sx should be renamed to "EV Table x" and the callouts need to be updated as well. Legends should be uploaded as a separate tab/sheet in each Excel file.
9. Appendix:
 - Appendix file needs to be in PDF format
 - Title page should contain "Appendix for [manuscript title]" and a Table of Content with the page numbers for the listed items;
 - nomenclature should be Appendix Figure Sx and Appendix Table Sx throughout the manuscript and Appendix PDF;
 - Supplementary Figure legends should be removed from the manuscript file.

10. Please provide a "standfirst text" summarizing the study in one or two sentences (approximately 250 characters, including space), three to four "bullet points" highlighting the main findings and a "synopsis image" (550px width and 400-600 px height, PNG format) to highlight the paper on our homepage.

Here are a couple of examples:

<https://www.embopress.org/doi/10.15252/msb.20199356>

<https://www.embopress.org/doi/10.15252/msb.20209475>

<https://www.embopress.org/doi/10.15252/msb.209495>

11. Please download and fill our Reagents and Tools Table template (.docx), which you can find in our author guidelines:

<https://www.embopress.org/page/journal/17444292/authorguide#structuredmethods>.

12. "Declaration of interests" should be renamed to "DISCLOSURE AND COMPETINGINTERESTS STATEMENT".

14. Data availability section- please use the following heading and format:

Data availability

The computer code produced in this study are available in the following database: <https://github.com/korcsmarosgroup/iSNP>

15. When you resubmit your manuscript, please download our CHECKLIST (<https://www.embopress.org/pb-assets/embo-site/EMBO%20Press%20Author%20Checklist-1642513524327.xlsx>) and include the completed form in your submission.

Please note that the Author Checklist will be published alongside the paper as part of the transparent process (<https://www.embopress.org/page/journal/17444292/authorguide#transparentprocess>).

16. "Materials and Methods" should be renamed to "Methods".

17. The manuscript requires the following section names and order: Title page - Abstract - Keywords - Introduction - Results - Discussion - Methods - Data Availability - Acknowledgements - Disclosure and Competing Interests Statement - References - Figure Legends - Table(s) - Expanded View Figure Legends.

If you feel you can satisfactorily deal with these points and those listed by the referees, you may wish to submit a revised version of your manuscript. Please attach a covering letter giving details of the way in which you have handled each of the points raised by the referees. A revised manuscript will be once again subject to review and you probably understand that we can give you no guarantee at this stage that the eventual outcome will be favorable.

Sincerely,
Jingyi

Jingyi Hou, PhD
Senior Editor
Molecular Systems Biology

*** PLEASE NOTE *** As part of the EMBO Press transparent editorial process initiative (see our Editorial at <https://dx.doi.org/10.1038/msb.2010.72>), Molecular Systems Biology publishes online a Review Process File with each accepted manuscripts. This file will be published in conjunction with your paper and will include the anonymous referee reports, your point-by-point response and all pertinent correspondence relating to the manuscript. If you do NOT want this File to be published, please inform the editorial office at contact@molsystbiol.org within 14 days upon receipt of the present letter.

Reviewer #4:

In the present work Módos et al. provided a substantially revised version of their manuscript, with improved method clarity, new figures, and additional text across different manuscript sections.

In particular, Reviewer 1's concerns (methods clarity, network bias, robustness of clustering, TF handling, figure clarity, DGE thresholds) were addressed with added details, new analyses, and revised figures.

Reviewer 2's concerns (rationale about mapping SNPs onto miRNA and TF-binding sites, experimental evidence, tissue specificity, GWAS power) were addressed with clarification, integration of experimental datasets, and expanded methods.

I strongly recommend the paper for publication.

There are, however, a few aspects that remain open:

Randomization analysis (Reviewer 1): the identification of several very highly studied TFs (e.g., TP53, NFKB1, STAT3) may be driven by such biases, and less-studied but more specific TFs may be missed. Degree-matched random seeds could be considered. Although I understand the authors performed several complementary analyses, I am not sure they compensated for the potential popularity bias (spread across the many chosen resources). They should consider using random seeds with the same degree as the input list (if this was already addressed, then I recommend to improve the clarity of the analysis which resulted obscure to me).

Presentation (Reviewer 1): please avoid the black background whenever possible; in Figure 4, Figure S3 (and many others) it is unnecessary and inconsistent with other figure presentations (e.g., Figure S4). I personally recommend adopting the black background exclusively for figures with graphs/networks.

Text in Fig. 2C, 7B, S8 (y-axes), S9 (right panels) is not visible.

Figure S7 lacks details.

Statistical rigor (Reviewer 2): the manuscript sometimes says "adj p value < 0.1" (Kong scRNA-seq) and elsewhere uses "adj p < 0.05" (pseudo-bulk/other datasets). The authors must state whether "p" = raw p or adjusted p (Benjamini-Hochberg FDR). They should also justify why they used a 0.1 threshold.

We thank the Editor and the Reviewer for kindly reviewing our revised manuscript. We have responded to the comments below:

Editorial comments

1. Reduce the keyword number to five.

We have now reduced the number of keywords to 5.

2. Remove the "Authors' contribution" section from the manuscript file.

We have now removed the Authors' contribution section from the manuscript file.

3. All figures should be removed from the main manuscript file. The figure legends need to be retained and placed in a separate section after the Reference list.

We have now removed the figures from the main manuscript file, and moved figure legends to a separate section after the Reference list.

4. On the title page, please clearly denote the corresponding author(s) within the author list.

We have now clearly denoted the corresponding author within the author list.

5. "Bibilography" should be renamed to "References".

We have now renamed "Bibliography" to "References"

6. Figure callouts: all callouts should be listed sequentially. Please add the missing callouts for Fig. 1A-E and 2C.

We have now added the missing callouts sequentially.

7. Please ensure the funding information in the manuscript matches the details provided in the submission system. The following grant is currently missing from the manuscript and must be added: UKRI | Biotechnology and Biological Sciences Research Council (BBSRC), grant number BBS/E/F/000PR10355.

Thank you for spotting this mistake. We have now added the missing grant to the manuscript.

8. For Tables S2-S4 and Tables S6-S8: Source file names, titles, legends and manuscript callouts all need to be updated to Dataset EV1-EV#, since they are rather large. The remaining Tables Sx should be renamed to "EV Table x" and the callouts need to be updated as well. Legends should be uploaded as a separate tab/sheet in each Excel file.

We have now renamed Tables S2-S4 and Tables S6-S8 as advised.

9. Appendix:

- Appendix file needs to be in PDF format
- Title page should contain "Appendix for [manuscript title]" and a Table of Content with the page numbers for the listed items;
- nomenclature should be Appendix Figure Sx and Appendix Table Sx throughout the manuscript and Appendix PDF;
- Supplementary Figure legends should be removed from the manuscript file.

We attached the Supplementary Figures as the editor asked.

10. Please provide a "standfirst text" summarizing the study in one or two sentences (approximately 250 characters, including space), three to four "bullet points" highlighting the main findings and a "synopsis image" (550px width and 400-600 px height, PNG format) to highlight the paper on our homepage.

Here are a couple of examples:

<https://www.embopress.org/doi/10.15252/msb.20199356>

<https://www.embopress.org/doi/10.15252/msb.20209475>

<https://www.embopress.org/doi/10.15252/msb.209495>

We have now provided a "standfirst text", "bullet points", and a "synopsis image" as advised.

11. Please download and fill our Reagents and Tools Table template (.docx), which you can find in our author

guidelines:<https://www.embopress.org/page/journal/17444292/authorguide#structuredmethods>.

We have now downloaded and filled out the Reagents and Tools Table.

12. "Declaration of interests" should be renamed to "DISCLOSURE AND COMPETING INTERESTS STATEMENT".

We have now renamed this as requested.

We have added the source data for the main manuscript figures.

14. Data availability section- please use the following heading and format:

Data availability

The computer code produced in this study are available in the following database:

<https://github.com/korcsmarosgroup/iSNP>

This has now been actioned.

15. When you resubmit your manuscript, please download our CHECKLIST

(<https://www.embopress.org/pb-assets/embo-site/EMBO%20Press%20Author%20Checklist-1642513524327.xlsx>) and include the completed form in your submission.

Please note that the Author Checklist will be published alongside the paper as part of the transparent process

(<https://www.embopress.org/page/journal/17444292/authorguide#transparentprocess>).

We have downloaded and completed the checklist.

16. "Materials and Methods" should be renamed to "Methods".

This has now been amended as requested.

17. The manuscript requires the following section names and order: Title page - Abstract - Keywords - Introduction - Results - Discussion - Methods - Data Availability - Acknowledgements - Disclosure and Competing Interests Statement - References - Figure Legends - Table(s) - Expanded View Figure Legends.

We have now followed this order of manuscript section names.

Reviewer #4:

- In the present work Módos et al. provided a substantially revised version of their manuscript, with improved method clarity, new figures, and additional text across different manuscript sections. In particular, Reviewer 1's concerns (methods clarity, network bias, robustness of clustering, TF handling, figure clarity, DGE thresholds) were addressed with added details, new analyses, and revised figures. Reviewer 2's concerns (rationale about mapping SNPs onto miRNA and TF-binding sites, experimental evidence, tissue specificity, GWAS power) were addressed with clarification, integration of experimental datasets, and expanded methods. I strongly recommend the paper for publication.

We are very thankful to the reviewer for their positive comments and for kindly acknowledging the substantial revision we undertook.

- There are, however, a few aspects that remain open:
Randomization analysis (Reviewer 1): the identification of several very highly studied TFs (e.g., TP53, NFKB1, STAT3) may be driven by such biases, and less-studied but more specific TFs may be missed. Degree-matched random seeds could be considered. Although I understand the authors performed several complementary analyses, I am not sure they compensated for the potential popularity bias (spread across the many chosen resources). They should consider using random seeds with the same degree as the input list (if this was already addressed, then I recommend to improve the clarity of the analysis which resulted obscure to me).

We are grateful for the Reviewer for this comment. While we had already undertaken the popularity bias and random seed degree-matched analyses, we agree with the Reviewer that this would have been obscure to the reader as they were described in the Methods section rather than in the main Results section. To address this, we have now added a section in the main Results entitled "Sensitivity and robustness analyses of network propagation results" encompassing these analyses.

- Presentation (Reviewer 1): please avoid the black background whenever possible; in Figure 4, Figure S3 (and many others) it is unnecessary and inconsistent with other figure presentations (e.g., Figure S4). I personally recommend adopting the black background exclusively for figures with graphs/networks. Text in Fig. 2C, 7B, S8 (y-axes), S9 (right panels) is not visible. Figure S7 lacks details.

We take on board the Reviewer's constructive feedback, and have amended the colour of the figure backgrounds accordingly, made the text clearer in the aforementioned figures, and added details to Appendix Figure S7.

- Statistical rigor (Reviewer 2): the manuscript sometimes says "adj p value < 0.1" (Kong scRNA-seq) and elsewhere uses "adj p < 0.05" (pseudo-bulk/other datasets). The authors must state whether "p" = raw p or adjusted p (Benjamini-Hochberg FDR). They should also justify why they used a 0.1 threshold.

We are sorry for these inconsistencies and thank the Reviewer to bringing it to our attention. We have now clarified that the adjusted p values were by the Benjamini-Hochberg method, and have also justified why in certain analyses a p < 0.1 significance threshold was selected. In the overrepresentation analysis of the Reactome pathways we used an empirical FDR of the MulEA package.

We aimed at a higher sensitivity at the pathway space to compensate for the high specificity of the network propagation analysis. This was our logic behind the permissive 0.1 cut-off.

We are grateful for the Reviewer for suggesting these changes as we believe these added value to the clarity and confidence of our analyses and their presentations.

27th Oct 2025

Manuscript number: MSB-2024-12721RR

Title: Decoding Non-coding SNPs: Systems Genomics Modelling Dissects the Heterogeneity of IBD

Dear Dr Korcsmaros,

Thank you again for sending us your revised manuscript. We are now satisfied with the modifications made and I am pleased to inform you that your paper has been accepted for publication.

Sincerely,
Jingyi

Jingyi Hou, PhD
Senior Editor
Molecular Systems Biology
